# On Learning to Solve Cardinality Constrained Combinatorial Optimization in One-Shot: A Re-parameterization Approach via Gumbel-Sinkhorn-TopK

## Abstract

Cardinality constrained combinatorial optimization requires selecting an optimal subset of $k$ elements, and it will be appealing to design data-driven algorithms that perform TopK selection over a probability distribution predicted by a neural network. However, the existing differentiable TopK operator suffers from an unbounded gap between the soft prediction and the discrete solution, leading to inaccurate estimation of the combinatorial objective score. In this paper, we present a self-supervised learning pipeline for cardinality constrained combinatorial optimization, which incorporates with Gumbel-Sinkhorn-TopK (GS-TopK) for near-discrete TopK predictions and the re-parameterization trick resolving the non-differentiable challenge. Theoretically, we characterize a bounded gap between the Maximum-A-Posteriori (MAP) inference and our proposed method, resolving the divergence issue in the previous differentiable TopK operator and also providing a more accurate estimation of the objective score given a provable tightened bound to the discrete decision variables. Experiments on max covering and discrete clustering problems show that under comparative time budget, our method outperforms state-of-the-art Gurobi solver and the novel one-shot learning method Erdos Goes Neural.

## 1 Introduction

In this paper, we aim to solve a family of cardinality constrained combinatorial optimization: given the cardinality budget $k$, our aim is to find the optimal set with size $k$ which minimizes the objective function $f(\mathbf{x})$. Denote $||\mathbf{x}||_0 \leq k$ as the cardinality constraint, we have:

$$\min_{\mathbf{x}} f(\mathbf{x}) \qquad s.t. \quad ||\mathbf{x}||_0 \leq k \tag{1}$$

Extensive applications of these problems can be found in machine learning and operations research, for example, discovering coresets for dataset summarization and continual learning (Borsos et al., 2020), discovering the most influential seed users in social networks (Chen et al., 2021), and planning facility locations in operation research (Liu, 2009). Although there has been a success in discovering the important submodularity (Fujishige, 1991) for a family of these problems with a bounded approximation ratio of greedy algorithms, the ubiquitous NP-hard challenge is always motivating us to develop improved algorithms with better efficiency-accuracy trade-off. The recent success of machine learning has inspired researchers to develop learning-based algorithms for combinatorial optimizations (Vinyals et al., 2015; Dai et al., 2016; Karalias & Loukas, 2020; Wang et al., 2021b), and this paper falls in line with these works.

According to the taxonomy by a recent survey (Peng et al., 2021), existing machine learning methods for combinatorial optimization can be mainly categorized into two groups: multi-step methods (Vinyals et al., 2015; Dai et al., 2016; Lu et al., 2019; Wang et al., 2021a) and one-shot methods (Wang et al., 2019; Karalias & Loukas, 2020; Li et al., 2019). The former line of works usually formulates the combinatorial optimization solving procedure as a Markov Decision Process (MDP) and refers to novel reinforcement learning methods (Mnih et al., 2013; Schulman et al., 2017) to tackle the MDP. Perhaps the largest drawback of such a multi-step pipeline is that the training and

inference procedures can be inefficient. Based on some previous efforts (Wang et al., 2019; Li et al., 2019), there is a recent seminal work (Karalias & Loukas, 2020) aiming to develop a one-shot pipeline for general combinatorial problems. Although being a fast and general pipeline, Karalias & Loukas (2020) do not encode constraint in the model prediction and thus the estimation of the combinatorial objective score may be inaccurate.

In this paper, we aim to facilitate the one-shot learning pipeline with cardinality constraints, such that the constraints can be softly encoded by the output of a neural network. Regarding the input to TopK as a probability distribution, the maximum-a-posterior (MAP) inference is done by sorting all probabilities and selecting the $K$-largest ones, however, this process is non-differentiable. Though appealing, it is non-trivial to encode the constraint in a differentiable manner with bounded gap w.r.t. the MAP inference and the bound of the existing SOFT-TopK algorithm (Xie et al., 2020) diverges if the $k$-th and $(k+1)$-th probabilities are equal. Unfortunately, such a dilemma is possible because the probabilities are outputs from a neural network. Besides, concerning the discrete decision variables in combinatorial optimization, it is also non-trivial to build an accurate estimation of the objective score with neural network output (which has to be continuous in order to preserve the gradient).

To tackle the non-differentiable challenge and the ill-estimated objective score, we refer to the seminal reparameterization technique with Gumbel noise, which can mimic the sampling procedure from the continuous probability distributions to near-discrete decision variables, while still preserving the gradient. Such a technique has been found successful for Top-1 (Jang et al., 2017), permutation (Mena et al., 2018), and sorting (Grover et al., 2019) distributions, and in this paper, we improve the SOFT-TopK operator (Xie et al., 2020) by introducing the Gumbel re-parameterization trick, and we theoretically prove that its gap can be bounded with the existence of the Gumbel noise, in contrast to the gap of SOFT-TopK (Xie et al., 2020) that may turn diverged.

To this end, we present **G**umbel-**S**inkhorn-**TopK** (GS-TopK) to facilitate the one-shot self-supervised learning pipeline (Karalias & Loukas, 2020) with constrained neural network output. The proposed GS-TopK operator improves the SOFT-TopK (Xie et al., 2020), where we theoretically characterize a bounded gap by introducing non-zero noise factors. Besides, we show how to integrate the proposed GS-TopK method into the combinatorial optimization pipeline, with an accurate estimation of the objective score, formulating a self-supervised learning pipeline following (Karalias & Loukas, 2020). The contributions of this paper are summarized as follows:

**1)** We propose a self-supervised learning approach to solve cardinality constrained combinatorial optimizations in one shot. The cardinality constraints are encoded in the model output via our proposed GS-TopK, and our pipeline is differentiable via the re-parameterization trick.
**2)** We theoretically characterize a bounded gap between the MAP inference and the approximate solution by Gumbel-Sinkhorn-TopK without requiring the large gap of $k$-th and $(k+1)$-th probabilities, which improves the result of the state-of-the-art SOFT-TopK algorithm (Xie et al., 2020). This property is crucial for an accurate estimation of the combinatorial objective score in learning.
**3)** Experiments on max covering and discrete clustering problems show that under a comparative or smaller inference time budget, our method surpasses the state-of-the-art one-shot learning methods e.g. (Karalias & Loukas, 2020), and also surpasses the state-of-the-art Gurobi and SCIP solvers.

## 2 RELATED WORK

### 2.1 LEARNING OF COMBINATORIAL OPTIMIZATION

Recently it is a trending topic of developing machine learning based combinatorial optimization methods to achieve a better trade-off between time and accuracy by modern data-driven approaches. Here we follow the taxonomy by Peng et al. (2021) and discuss some representative works.

**Multi-step methods** predict the solution by a multi-step decision process. Since the fundamental work (Vinyals et al., 2015) that proposes an encoding-decoding architecture, researchers have explored this general pipeline to various real-world combinatorial optimization problems. The most popular framework follows the reinforcement learning pipeline introduced by Khalil et al. (2017), where the state is encoded by a graph neural network, and multi-step actions are taken to sequentially construct the solution. Applications have been explored on job scheduling (Zhang et al., 2020), computer resource allocation (Mao et al., 2019), quadratic assignment programming (Liu et al., 2020), bin-packing (Duan et al., 2019), just to name a few. The advantage of multi-step methods is that they can handle combinatorial constraints that are non-trivial to be encoded in a differentiable manner, by

adding constraints on the feasible actions in the multi-step roll-out procedure. However, the training of reinforcement learning may be inefficient and vulnerable, and there are also some recent efforts on developing one-shot combinatorial optimization learning methods.

**One-shot methods** predict the solution by a single forward pass. Despite the challenges in encoding general constraints, one-shot learning methods have been successful on certain problems where the constraint can be encoded by the neural network. For example, in graph matching (Wang et al., 2019; 2021b) the constraint can be encoded by Sinkhorn algorithm (Sinkhorn & Rangarajan, 1964), and also for graph similarity regression problems (Bai et al., 2019; Li et al., 2019) that are unconstrained. The seminal work (Karalias & Loukas, 2020) aims to develop a general pipeline for one-shot learning of combinatorial optimization, by encoding the violation of the constraint as part of its objective score. However, our experimental results show that compared to Karalias & Loukas (2020), it will be more appealing if we can explicitly encode the constraint in the neural network output.

### 2.2 Optimal Transport and Gumbel Re-parameterization

**Optimal Transport.** The optimal transport (OT) is a linear programming problem that aims to find the optimal transportation plan with minimal cost. Though the development of the Sinkhorn algorithm can be date back to 1960s (Sinkhorn & Rangarajan, 1964), it has drawn increasing attention from the machine learning community since the pioneering work (Cuturi, 2013) which introduces the entropic-regularization for the optimal transport problem. Since the Sinkhorn algorithm (Sinkhorn & Rangarajan, 1964) incorporates row- and column-wise normalization which is differentiable, OT can be incorporated within any end-to-end deep learning pipelines. Thus, OT has been found successful for deep generative models (Arjovsky et al., 2017; Patrini et al., 2019), solving jigsaw puzzles (Santa Cruz et al., 2018), and deep graph matching (Fey et al., 2020; Yu et al., 2020). In complement to the recent line of differentiable sorting papers (Petersen et al., 2021; Cuturi et al., 2019), Xie et al. (2020) develop the differentiable TopK algorithm namely SOFT-TopK, by formulating the TopK problem as an OT problem. However, the theoretical gap between the prediction of SOFT-TopK and the result obtained by sorting diverges if the $k$-th and $(k+1)$-th largest items are equal. In this paper, we prove that this gap can be bounded by introducing Gumbel noise.

**Gumbel Re-parameterization.** The Gumbel distribution with the re-parameterization trick is shown effective to mimic the sampling procedure from a given distribution in a differentiable manner. Jang et al. (2017); Maddison et al. (2017) take the initiative to develop Gumbel-Softmax, where one-hot vectors are sampled from a softmax probability distribution. The non-differentiable sampling procedure turned to be differentiable by the namely re-parameterization trick (Kingma & Welling, 2013). Inspired by Jang et al. (2017), Mena et al. (2018) propose a Gumbel-Sinkhorn method to learn the latent permutations, which is relevant to the GS-TopK method developed in this paper. Besides, Paulus et al. (2020) develop a general framework for generating combinatorial distributions via random distributions. However, these existing methods do not study the application in combinatorial optimization learning, and the gap between the MAP inference and the result after Gumbel re-parameterization is not well characterized in Mena et al. (2018). In this paper, we explore the application of the re-parameterization trick with Gumbel noise in combinatorial optimization, also we theoretically characterize the gap between the MAP inference and the proposed Gumbel method.

## 3 The Proposed Gumbel-Sinkhorn-TopK

### 3.1 Preliminary: SOFT-TopK Algorithm

The TopK problems are ubiquitous in machine learning, yet the MAP inference approach (performing sorting at the time cost of $\mathcal{O}(m \log m)$ and selecting the first $k$ items) is unfriendly for current gradient-based deep neural networks. To introduce differentiable TopK for machine learning, Xie et al. (2020) formulate the TopK problem as an optimal transport (OT) problem. Specifically, for selecting the $k$ largest items from the probability vector $\mathbf{s}$ of size $m$ ($k < m$), the OT problem refers to moving $k$ items to one destination (selected), and the other $m - k$ elements to another destination (not selected). Thus, the marginal distributions of the OT are:

$$\mathbf{c} = \underbrace{[1 \quad 1 \quad ... \quad 1]}_{m \text{ items}} \qquad \mathbf{r} = \left[ \begin{array}{c} m - k \\ k \end{array} \right] \Big\} 2 \text{ destinations} \qquad (2)$$

---

**Algorithm 1: Gumbel-Sinkhorn-TopK (GS-TopK) for Solving Cardinality Constrained Combinatorial Optimization**

---

**Input:** List $\mathbf{s}$ with $m$ items; TopK size $k$; Sinkhorn factor $\tau$; noise factor $\sigma$; sample size #G.

1  **for** $a \in \{1, 2, ..., \#G\}$ **do**

2      for all $\mathbf{s}_i$, $\widetilde{\mathbf{s}}_i = \mathbf{s}_i - \sigma \log(-\log(u_i))$, where $u_i$ is sampled from $(0, 1)$ uniform distribution;

3      $\widetilde{\mathbf{D}} = \begin{bmatrix} \widetilde{\mathbf{s}}_1 - \min(\mathbf{s}) & ... & \widetilde{\mathbf{s}}_m - \min(\mathbf{s}) \\ \max(\mathbf{s}) - \widetilde{\mathbf{s}}_1 & ... & \max(\mathbf{s}) - \widetilde{\mathbf{s}}_m \end{bmatrix}$; $\mathbf{c} = \underbrace{[1 \quad 1 \quad ... \quad 1]}_{m \text{ items}}$; $\mathbf{r} = \begin{bmatrix} m - k \\ k \end{bmatrix}$;

4      $\widetilde{\mathbf{T}}_a = \exp(-\widetilde{\mathbf{D}}/\tau)$;

5      **while** *not converged* **do**

6         $\widetilde{\mathbf{D}}_r = \mathrm{diag}(\widetilde{\mathbf{T}}_a \mathbf{1} \oslash \mathbf{r})$;    $\widetilde{\mathbf{T}}_a = \widetilde{\mathbf{D}}_r^{-1} \widetilde{\mathbf{T}}_a$;

7         $\widetilde{\mathbf{D}}_c = \mathrm{diag}(\widetilde{\mathbf{T}}_a^\top \mathbf{1} \oslash \mathbf{c})$;    $\widetilde{\mathbf{T}}_a = \widetilde{\mathbf{T}}_a \widetilde{\mathbf{D}}_c^{-1}$;

**Output:** A list of transport matrices $[\widetilde{\mathbf{T}}_1, \widetilde{\mathbf{T}}_2, ..., \widetilde{\mathbf{T}}_{\#G}]$.

---

We can design the destinations to be the min/max values of $\mathbf{s}$, such that the TopK items are moved to $\max(\mathbf{s})$, and the other items are moved to $\min(\mathbf{s})$. Then, the distance matrix of OT is given as:

$$\mathbf{D} = \begin{bmatrix} \mathbf{s}_1 - \min(\mathbf{s}) & \mathbf{s}_2 - \min(\mathbf{s}) & ... & \mathbf{s}_m - \min(\mathbf{s}) \\ \max(\mathbf{s}) - \mathbf{s}_1 & \max(\mathbf{s}) - \mathbf{s}_2 & ... & \max(\mathbf{s}) - \mathbf{s}_m \end{bmatrix}. \tag{3}$$

While the OT is formulated as an integer linear programming problem:

$$\min_{\mathbf{T}} \mathrm{tr}(\mathbf{T}^\top \mathbf{D}) \qquad s.t. \quad \mathbf{T} \in \{0, 1\}^{m \times 2}, \mathbf{T}\mathbf{1} = \mathbf{r}, \mathbf{T}^\top \mathbf{1} = \mathbf{c}, \tag{4}$$

where $\mathbf{T}$ is the transportation plan which also corresponds to the TopK solution of the problem, and $\mathbf{1}$ is a column vector whose all elements are 1s. The optimal solution $\mathbf{T}^*$ should be equivalent to the solution by firstly sorting all items and then selecting the TopK items. It is also equivalent to the MAP inference by regarding $\mathbf{s}_1, ..., \mathbf{s}_m$ as probabilities. Following the differentiable Sinkhorn algorithm for OT (Cuturi, 2013; Sinkhorn & Rangarajan, 1964), the binary constraint on $\mathbf{T}$ is relaxed to continuous values $[0, 1]$, and Eq. (4) is modified with entropic regularization:

$$\min_{\mathbf{T}^\tau} \mathrm{tr}(\mathbf{T}^{\tau\top} \mathbf{D}) + \tau h(\mathbf{T}^\tau) \qquad s.t. \quad \mathbf{T}^\tau \in [0, 1]^{m \times 2}, \mathbf{T}^\tau \mathbf{1} = \mathbf{r}, \mathbf{T}^{\tau\top} \mathbf{1} = \mathbf{c}, \tag{5}$$

where $h(\mathbf{T}^\tau) = \sum_{i,j} \mathbf{T}_{ij}^\tau \log \mathbf{T}_{ij}^\tau$ is the entropic regularizer (Cuturi, 2013). Given any real-valued matrix $\mathbf{D}$, Eq. (5) is solved by firstly enforcing the regularization factor $\tau$: $\mathbf{T}^\tau = \exp(-\mathbf{D}/\tau)$. Then $\mathbf{T}^\tau$ is row- and column-wise normalized alternatively:

$$\mathbf{D}_r = \mathrm{diag}(\mathbf{T}^\tau \mathbf{1} \oslash \mathbf{r}), \quad \mathbf{T}^\tau = \mathbf{D}_r^{-1} \mathbf{T}^\tau, \quad \mathbf{D}_c = \mathrm{diag}(\mathbf{T}^{\tau\top} \mathbf{1} \oslash \mathbf{c}), \quad \mathbf{T}^\tau = \mathbf{T}^\tau \mathbf{D}_c^{-1}, \tag{6}$$

where $\oslash$ means element-wise division. We denote $\mathbf{T}^{\tau*}$ as the converged solution, which is the optimal solution to Eq. (5).

**Theorem 2 of (Xie et al., 2020).** Denote $x_k, x_{k+1}$ as the $k$-th $(k + 1)$-th largest items, we have

$$||\mathbf{T}^* - \mathbf{T}^{\tau*}||_F \leq \frac{2m\tau \log 2}{|x_k - x_{k+1}|}, \tag{7}$$

where the gap is controlled by $\tau$ if $|x_k - x_{k+1}| > 0$. However, the above bound diverges if $x_k = x_{k+1}$, which is unavoidable because $x_k, x_{k+1}$ are outputs by a neural network. It motivates us to develop our improved version Gumbel-Sinkhorn-TopK by introducing the Gumbel noise.

## 3.2 GUMBEL-SINKHORN-TOPK (GS-TOPK) ALGORITHM

In this section, we present our proposed **G**umbel-**S**inkhorn-**TopK** (GS-TopK) as summarized in Alg. 1 and we theoretically characterize the gap between the MAP inference and the prediction by GS-TopK. As the re-parameterization trick (Jang et al., 2017), instead of sampling from a distribution that is non-differentiable, we add random variables to probabilities predicted by neural networks. The Gumbel distribution is characterized as:

$$g_\sigma(u) = -\sigma \log(-\log(u)), \tag{8}$$

where $\sigma$ controls the variance and $u$ is from $(0, 1)$ uniform distribution. We can update $\mathbf{s}$ and $\mathbf{D}$ as:

$$\widetilde{\mathbf{s}}_i = \mathbf{s}_i + g_\sigma(u_i) \tag{9}$$

$$\widetilde{\mathbf{D}} = \left[ \begin{array}{cccc} \widetilde{\mathbf{s}}_1 - \min(\mathbf{s}) & \widetilde{\mathbf{s}}_2 - \min(\mathbf{s}) & ... & \widetilde{\mathbf{s}}_m - \min(\mathbf{s}) \\ \max(\mathbf{s}) - \widetilde{\mathbf{s}}_1 & \max(\mathbf{s}) - \widetilde{\mathbf{s}}_2 & ... & \max(\mathbf{s}) - \widetilde{\mathbf{s}}_m \end{array} \right]. \tag{10}$$

Again we formulate the integer linear programming version of the OT with Gumbel noise:

$$\min_{\mathbf{T}^\sigma} \mathrm{tr}(\mathbf{T}^{\sigma\top}\widetilde{\mathbf{D}}) \qquad s.t. \quad \mathbf{T}^\sigma \in \{0,1\}^{m\times 2}, \mathbf{T}^\sigma\mathbf{1} = \mathbf{r}, \mathbf{T}^{\sigma\top}\mathbf{1} = \mathbf{c}, \tag{11}$$

where the optimal solution to Eq. (11) is denoted as $\mathbf{T}^{\sigma*}$. Without loss of generality, in the following we sort $\mathbf{s}_1, \mathbf{s}_2, \mathbf{s}_3, ..., \mathbf{s}_m$ in descending order, into $\mathcal{X} = x_1, x_2, x_3, ..., x_m$, and $x_k, x_{k+1}$ are the $k$-th and $(k + 1)$-th largest items, respectively. We characterize the gap between $\mathbf{T}^*$ and $\mathbf{T}^{\sigma*}$ under the expectation over Gumbel distribution.

Now we present the following two lemmas to help establish our theoretical results, in terms of a bounded gap between the MAP inference and the approximate solution by Gumbel-Sinkhorn-TopK,

**Lemma 1 (Bias by the Gumbel Noise).** *For the integer linear programming solutions with and without Gumbel noises, we have*

$$\mathbb{E}_u\left[||\mathbf{T}^* - \mathbf{T}^{\sigma*}||_F\right] \leq \sqrt{\frac{4m}{1 + \exp\frac{|x_k - x_{k+1}|}{\sigma}}} \tag{12}$$

*Proof sketch:* The gap between $\mathbf{T}^*, \mathbf{T}^{\sigma*}$ is derived by counting the expected number of times that, after adding the Gumbel noise, an element may become larger than $x_k$ or smaller than $x_{k+1}$. For the top $k$ elements, the probability that it will become smaller than $x_{k+1}$ is upper bounded by $1/(1 + \exp\frac{|x_k - x_{k+1}|}{\sigma})$. For the last $(m-k)$ elements, the probability that it will become larger than $x_k$ is also upper bounded by $1/(1 + \exp\frac{|x_k - x_{k+1}|}{\sigma})$. Since $||\mathbf{T}^* - \mathbf{T}^{\sigma*}||_F^2$ changes at most by 4 if an item becomes larger/smaller than $x_k/x_{k+1}$, and we have $m$ items, then we can proof the upper bound of this gap. The detailed proof is referred to Appendix A.1. $\qquad\square$

To make the integer linear programming problem feasible for gradient-based deep learning methods, we also relax the integer constraint and add the entropic regularization term:

$$\min_{\widetilde{\mathbf{T}}} \mathrm{tr}(\widetilde{\mathbf{T}}^\top\widetilde{\mathbf{D}}) + h(\widetilde{\mathbf{T}}) \qquad s.t. \quad \widetilde{\mathbf{T}} \in [0,1]^{m\times 2}, \widetilde{\mathbf{T}}\mathbf{1} = \mathbf{r}, \widetilde{\mathbf{T}}^\top\mathbf{1} = \mathbf{c}, \tag{13}$$

which is solved by Sinkhorn algorithm on $\widetilde{\mathbf{D}}$: firstly $\widetilde{\mathbf{T}} = \exp(-\widetilde{\mathbf{D}}/\tau)$, then

$$\widetilde{\mathbf{D}}_r = \mathrm{diag}(\widetilde{\mathbf{T}}\mathbf{1} \oslash \mathbf{r}), \quad \widetilde{\mathbf{T}} = \widetilde{\mathbf{D}}_r^{-1}\widetilde{\mathbf{T}}, \quad \widetilde{\mathbf{D}}_c = \mathrm{diag}(\widetilde{\mathbf{T}}^\top\mathbf{1} \oslash \mathbf{c}), \quad \widetilde{\mathbf{T}} = \widetilde{\mathbf{T}}\widetilde{\mathbf{D}}_c^{-1}. \tag{14}$$

Here we denote the optimal solution to Eq. (13) as $\widetilde{\mathbf{T}}^*$. We theoretically characterize the gap between $\mathbf{T}^{\sigma*}$ and $\widetilde{\mathbf{T}}^*$, which corresponds to the optimal solutions of Eq. (11) and Eq. (13), respectively.

**Lemma 2 (Gap to a Discrete Solution).** *Under probability $(1 - \epsilon)$, we have*

$$\mathbb{E}_u\left[||\mathbf{T}^{\sigma*} - \widetilde{\mathbf{T}}^*||_F\right] \leq (\log 2)m\tau \sum_{i\neq j} \Omega(x_i, x_j, \sigma, \epsilon), \tag{15}$$

*where*

$$\Omega(x_i, x_j, \sigma, \epsilon) = \frac{2\sigma\log\left(\sigma - \frac{|x_i - x_j| + 2\sigma}{\log(1-\epsilon)}\right) + |x_i - x_j|\left(\frac{\pi}{2} + \arctan\frac{x_i - x_j}{2\sigma}\right)}{(1-\epsilon)((x_i - x_j)^2 + 4\sigma^2)(1 + \exp\frac{x_i - x_k}{\sigma})(1 + \exp\frac{x_{k+1} - x_j}{\sigma})}, \tag{16}$$

*Proof sketch:* The gap between $\mathbf{T}^{\sigma*}$ and $\widetilde{\mathbf{T}}^*$ is a generalization from the Theorem 2 of Xie et al. (2020). We denote $x_{\pi_k}, x_{\pi_{k+1}}$ as the $k$-th and $(k + 1)$-th largest items after disturbed by the Gumbel noise, and our aim becomes to prove the upper bound of $\mathbb{E}_u\left[1/(|x_{\pi_k} + g_\sigma(u_{\pi_k}) - x_{\pi_{k+1}} - g_\sigma(u_{\pi_{k+1}})|)\right]$, where the probability density function of $g_\sigma(u_{\pi_k}) - g_\sigma(u_{\pi_{k+1}})$ can be bounded by $f(y) = 1/(y^2 + 4)$. Thus we can compute the upper bound under probability $(1 - \epsilon)$ by integration. The detailed proof is referred to Appendix A.2. $\quad\square$

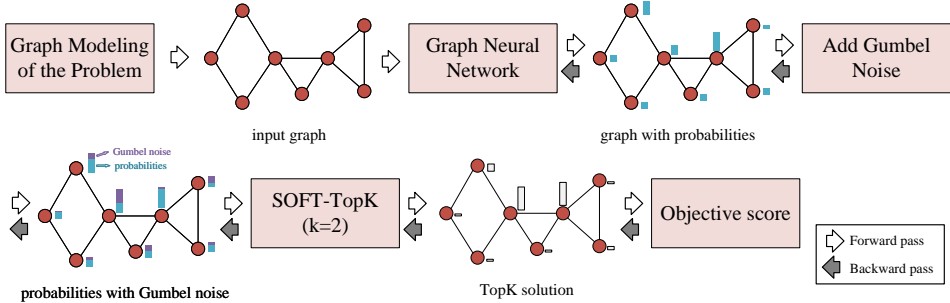

Figure 2: Our proposed self-supervised learning pipeline for combinatorial optimization. The probability of selecting each element is predicted by a graph neural network, and an accurate estimation of the combinatorial objective score is achieved via near-discrete TopK solution by our GS-TopK.

**Theorem 1.** *For the proposed Gumbel TopK solver, denote* $\mathbf{T}^*$ *as the solution by deterministic TopK algorithm,* $\widetilde{\mathbf{T}}^*$ *as the solution of Gumbel-Sinkhorn-TopK. With probability* $(1 - \epsilon)$*, we have*

$$\mathbb{E}_u \left[ \left\| \mathbf{T}^* - \widetilde{\mathbf{T}}^* \right\|_F \right] \leq \sqrt{\frac{4m}{1 + \exp\frac{|x_k - x_{k+1}|}{\sigma}}} + (\log 2)m\tau \sum_{i \neq j} \Omega(x_i, x_j, \sigma, \epsilon), \qquad (17)$$

*where* $\sigma$ *is the noise factor and* $\tau$ *is the temperature of Sinkhorn algorithm.* $m$ *is the number of candidates to be selected.* $\Omega(x_i, x_j, \sigma, \epsilon)$ *is defined in Eq. (16).*

*Proof:* Based on **Lemma 1** and **Lemma 2**, **Theorem 1** is proved by triangle inequality:

$$\mathbb{E}_u \left[ \left\| \mathbf{T}^* - \widetilde{\mathbf{T}}^* \right\|_F \right] \leq \mathbb{E}_u \left[ \|\mathbf{T}^* - \mathbf{T}^{\sigma*}\|_F \right] + \mathbb{E}_u \left[ \|\mathbf{T}^{\sigma*} - \widetilde{\mathbf{T}}^*\|_F \right] \qquad \square$$

**Remarks**. As derived above, the gap between $\mathbf{T}^*$ and $\widetilde{\mathbf{T}}$ is composed of two parts: the bias by the Gumbel noise (in Lemma 1) and the gap to a discrete solution (in Lemma 2). Interestingly, the first term becomes smaller given a smaller $\sigma$, and the second term is smaller given a larger $\sigma$ or a smaller $\tau$. See the toy example in Fig. 1, a larger $\sigma$ can tighten the gap of GS-TopK to a discrete solution, which is welcomed in our combinatorial optimization learning scenario where $\widetilde{\mathbf{T}}^*$ is used to compute the objective score in a self-supervised learning pipeline.

If $|x_k - x_{k+1}| > 0$, with $\sigma \to 0^+$, Eq. (17) degenerates to the bound derived by (Xie et al., 2020) and only differs by a constant factor (see Appendix A.3 for details):

$$\lim_{\sigma \to 0^+} \mathbb{E}_u \left[ \|\mathbf{T}^* - \widetilde{\mathbf{T}}^*\|_F \right] \leq \frac{(\pi \log 2)m\tau}{(1 - \epsilon)|x_k - x_{k+1}|}$$

It makes a strong assumption that $|x_k - x_{k+1}| > 0$, and the bound diverges if $x_k = x_{k+1}$. Un-

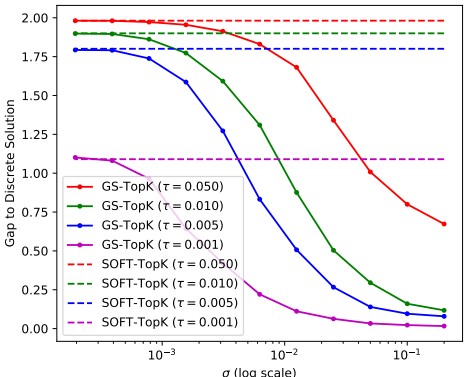

Figure 1: A toy example to explain Lemma 2: select top-3 items from $[1.0, 0.8, 0.601, 0.6, 0.4, 0.2]$ and we compare GS-TopK and SOFT-TopK concerning the gap to a discrete solution w.r.t. different $\tau, \sigma$ configurations. Here the gap to discrete solutions is tighten by a larger $\sigma$ and a smaller $\tau$ for GS-TopK, compared to SOFT-TopK whose gap is larger and can only be controlled by $\tau$.

fortunately, $x_k, x_{k+1}$ are predictions by a neural network, whereby such an assumption may not be satisfied. In comparison, given $\sigma > 0$, our conclusion in Eq. (17) is bounded for any $x_k, x_{k+1}$.

### 3.3 APPLICATION TO SELF-SUPERVISED COMBINATORIAL OPTIMIZATION LEARNING

In the following, we discuss the application of GS-TopK to self-supervised combinatorial optimization learning, by taking the max covering problem as an example. We consider the max covering problem with $m$ sets and $n$ objects. Each set may cover any number of objects, and each object is

---

**Algorithm 2: Gumbel-Sinkhorn-TopK Learning for Solving the Max Covering Problem**

---

**Input:** bipartite adjacency $\mathbf{A}$; values $\mathbf{v}$; learning rate $\alpha$; GS-TopK parameters $k, \tau, \sigma$, #G.

1 **if** *Training* **then**
2     Randomly initialize neural network weights $\theta$;
3 **if** *Inference* **then**
4     Load a pretrained neural network weights $\theta$; $J_{best} = 0$;
5 **while** *not converged* **do**
6     $\mathbf{s} = \text{GraphSage}_\theta(\mathbf{A})$; $[\widetilde{\mathbf{T}}_1, \widetilde{\mathbf{T}}_2, ..., \widetilde{\mathbf{T}}_{\#G}] = \text{GS-TopK}(\mathbf{s}, k, \tau, \sigma, \#G)$;
7     for all $i$, $\widetilde{J}_i = \min(\widetilde{\mathbf{T}}_i[2, :]\mathbf{A}, 1) \cdot \mathbf{v}$; $J = \text{mean}([\widetilde{J}_1, \widetilde{J}_2, ..., \widetilde{J}_{\#G}])$;
8     **if** *Training* **then**
9        update $\theta$ with respect to the gradient $\frac{\partial J}{\partial \theta}$ and learning rate $\alpha$ by gradient ascent;
10     **if** *Inference* **then**
11        update $\mathbf{s}$ with respect to the gradient $\frac{\partial J}{\partial \mathbf{s}}$ and learning rate $\alpha$ by gradient ascent;
12        for all $i$, $\widetilde{J}_i = \text{TopK}(\widetilde{\mathbf{T}}_i[2, :])\mathbf{A} \cdot \mathbf{v}$; $J_{best} = \max([\widetilde{J}_1, \widetilde{J}_2, ..., \widetilde{J}_{\#G}], J_{best})$;
13 **if** *Homotopy Inference* **then**
14     Shrink the values of $\tau, \sigma$ and jump to line 5;

**Output:** Learned network weights $\theta$ (if training)/The best objective $J_{best}$ (if inference).

---

associated with a value. We aim to find $k$ sets (where $k < m$) such that the covered objects have the maximum sum of values. The general illustration of our pipeline can be found in Fig. 2.

Firstly, we build a bipartite graph whose two disjoint sets of vertices are the sets and the objects. An edge is defined if an object is covered by a set. Denote $\mathbf{v} \in \mathbb{R}^n$ as the value of each object, and $\mathbf{A} \in \{0, 1\}^{m \times n}$ as the adjacency matrix, the problem is formulated as

$$\max_{\mathbf{x}} \sum_{j=1}^{n} \left( \mathbb{I}\left(\sum_{i=1}^{m} \mathbf{x}_i \mathbf{A}_{ij}\right) \cdot \mathbf{v}_j \right) \qquad s.t. \quad \mathbf{x} \in \{0, 1\}^m, ||\mathbf{x}||_0 \leq k \qquad (18)$$

where $\mathbb{I}(\mathbf{x})$ is an indicator $\mathbb{I}(\mathbf{x})_i = 1$ if $\mathbf{x}_i \geq 1$ else $\mathbb{I}(\mathbf{x})_i = 0$. To encode the bipartite graph, we exploit three layers of GraphSage (Hamilton et al., 2017) followed by a fully-connected layer with sigmoid to predict the probability of selecting each set, which is denoted as $\mathbf{s} \in [0, 1]^m$. The probabilities $\mathbf{s}$ are fed into the GS-TopK algorithm, which outputs a batch of near-discrete transportation matrices $[\widetilde{\mathbf{T}}_1, \widetilde{\mathbf{T}}_2, ..., \widetilde{\mathbf{T}}_{\#G}]$, whose second row is regarded as a continuous approximation of the decision variables. The objective value is estimated as:

$$\widetilde{J}_i = \min(\widetilde{\mathbf{T}}_i[2, :]\mathbf{A}, 1) \cdot \mathbf{v}, \qquad J = \text{mean}([\widetilde{J}_1, \widetilde{J}_2, ..., \widetilde{J}_{\#G}]) \qquad (19)$$

For the discrete clustering problem, we aim to select $k$ objects from the full set of $m$ objects, by minimizing the sum of distance for each object between itself and its nearest selected object. It differs from the k-means clustering by adding the "discrete" constraint that cluster centers have to be a subset of objects, which is also known as facility location problem in operations research (Liu, 2009). Denote $\mathbf{\Delta} \in \mathbb{R}^{+m \times m}$ as the distance matrix for each pair of objects, we formulate the discrete clustering problem as follows (please refer to Appendix B for details):

$$\min_{\mathbf{x}} \sum_{j=1}^{m} \min(\{\mathbf{\Delta}_{i,j} | \forall \mathbf{x}_i = 1\}) \qquad s.t. \quad \mathbf{x} \in \{0, 1\}^m, ||x||_0 \leq k \qquad (20)$$

As illustrated in Alg. 2, during training, the Adam optimizer (Kingma & Ba, 2014) is applied for learning the neural network weights. During inference, the neural network prediction is regarded as initialization, and we again optimize the distribution w.r.t. the combinatorial objective score by Adam, during which a search procedure is also performed among all Gumbel samples.

**Homotopy GS-TopK.** Based on our Theorem 1, the gap between $\mathbf{T}^*$ and $\widetilde{\mathbf{T}}^*$ can be tightened by shrinking $\tau$ and $\sigma$, which motivates us to develop a homotopy variant (Xiao & Zhang, 2012; Xu et al., 2016) of GS-TopK with gradually decreased $\tau, \sigma$, such that $\widetilde{\mathbf{T}}^*$ and $\widetilde{J}$ can provide more accurate estimation during inference. See line 13 in Alg. 2 for details.

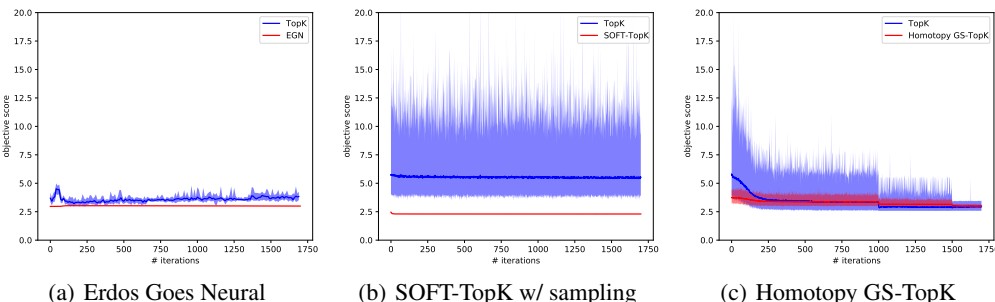

Figure 3: In discrete clustering, comparison of the estimated objective score (red) and the real one (blue). Our GS-TopK achieves the best accurate estimation of the objective score.

# 4 EXPERIMENTS AND DISCUSSION

## 4.1 PROTOCOLS AND BASELINES

We apply the proposed GS-TopK algorithm to learning two representative problems of cardinality constrained NP-hard combinatorial optimization: i) max covering and ii) discrete clustering problem. For each problem, both a smaller-sized case and a larger-sized case are considered, and we build separate training/testing datasets for learning methods. In this paper, we referred to randomly generated datasets following previous machine learning for combinatorial optimization papers (Khalil et al., 2017; Kool et al., 2019) concerning the limited large-scale open-source dataset. For the max covering problem, our data distribution follows the distribution in ORLIB[1]. For the discrete clustering problem, the data points are uniformly generated in a 2D plane.

We compare the following representative and diverse baselines with our proposed method:

**Classic Algorithms**. Classic algorithms are well-developed strong baselines with theoretical guarantees. In our experiment, we consider the greedy algorithm as the baseline for the max covering problem, which has the worst-case approximation ratio of $(1 - 1/e)$ due to the submodularity of the max covering problem (Fujishige, 1991). For the discrete clustering problem, we resort to the classic kmeans (Macqueen, 1967) and kmeans++ (Arthur & Vassilvitskii, 2007) with improved initialization techniques. These algorithms are fast but effective, even surpassing commercial solvers with only $1/1000$ inference time on certain test cases.

**General Purpose Solvers**. We also compare two general-purpose solvers Gurobi 9.0 (under educational license) (Gurobi Optimization, LLC, 2021) and SCIP 7.0 (Gamrath et al., 2020). Gurobi is the state-of-the-art commercial solver for general combinatorial optimization tasks, and it is known heavily engineered by various speedup techniques. SCIP is the state-of-the-art open-source solver, which is also regarded as a baseline. We formulate the combinatorial problems as integer programming and then call the solver to solve the problems, with a limited time budget. Our implementation of the solvers is with the Google ORTools API [2].

**One-shot Learning Baselines**. We compare with the general Erdos Goes Neural (EGN) (Karalias & Loukas, 2020) framework that models each decision variable as a Bernoulli distribution and encodes the constraint as part of the learning objective. Since the authors of (Karalias & Loukas, 2020) do not consider cardinality constrained problems in their experiment, we re-implement and tune the EGN model based on their official implementation[3]. We also compare with SOFT-TopK (Xie et al., 2020), which is theoretically characterized as a special case of our proposed GS-TopK when $\sigma = 0$. Also to ensure the effectiveness of the re-parameterization trick, we facilitate SOFT-TopK with Gumbel sampling during inference. All one-shot learning methods are with the same model structure for a fair comparison and are trained and tested on separate datasets. Besides, we empirically find the learning process of one-shot methods converges within tens of minutes. Since the reinforcement learning methods (Khalil et al., 2017; Chen & Tian, 2019) require significantly more training time, we only include one-shot methods for a fair comparison.

---

[1] http://people.brunel.ac.uk/~mastjjb/jeb/orlib/scpinfo.html
[2] https://developers.google.com/optimization
[3] https://github.com/Stalence/erdos_neu

Table 1: Objective score ↑ and inference time (in seconds) ↓ comparison of the max covering problem, including mean and standard deviation. Under cardinality constraint $k$, the problem is to select from $m$ sets to cover a fraction of $n$ objects. Gurobi solver fails to return the optimal solution within 24 hours, thus reported as out-of-time. Our GS-TopK outperforms all competing methods, with comparable or smaller time costs. The performance is further improved by Homotopy GS-TopK.

| EGN/SOTF-TopK/ours are one-shot solvers | k=50, m=500, n=1000 | | k=100, m=1000, n=2000 | |
|---|---|---|---|---|
| | objective ↑ | time ↓ (sec) | objective ↑ | time ↓ (sec) |
| greedy | 44312.8±818.4 | **0.024±0.000** | 88698.9±1217.5 | **0.089±0.001** |
| SCIP 7.0 (faster) | 43034.7±869.2 | 30.058±0.017 | 86269.9±1256.3 | 59.916±1.752 |
| SCIP 7.0 (slower) | 43497.4±875.6 | 100.136±0.097 | 86269.9±1256.3 | 120.105±0.498 |
| Gurobi 9.0 (faster) | 43261.6±856.1 | 30.078±0.014 | 85992.4±1643.7 | 60.168±0.042 |
| Gurobi 9.0 (slower) | 43937.2±791.5 | 100.171±0.085 | 86862.1±1630.5 | 120.277±0.139 |
| Gurobi 9.0 (optimal) | OOT | OOT | OOT | OOT |
| EGN (efficient) (Karalias & Loukas, 2020) | 36423.7±1128.4 | 0.244±0.107 | 70336.7±1676.9 | 0.525±0.229 |
| EGN (accurate) (Karalias & Loukas, 2020) | 36927.3±1163.0 | 40.542±4.056 | 71487.5±1552.8 | 93.670±8.797 |
| SOFT-TopK (Xie et al., 2020) | 41959.9±803.4 | 48.605±5.783 | 83211.7±1322.3 | 45.252±0.713 |
| SOFT-TopK (w/ sampling) | 39684.8±675.6 | 44.960±4.390 | 77285.2±1123.9 | 59.488±0.143 |
| GS-TopK (ours) | 44710.3±770.9 | 32.839±3.227 | 89264.8±1232.1 | 60.685±0.045 |
| Homotopy GS-TopK (ours) | **44718.2±745.2** | 47.627±4.247 | **89294.3±1211.2** | 89.764±0.128 |

Table 2: Objective score ↓, optimal gap ↓ and inference time (in seconds) ↓ comparison of the discrete clustering problem, with mean and standard deviation. The problem is to select $k$ clustering centers from $m$ data points. GS-TopK and Homotopy GS-TopK surpasses all competing methods with less time compared to general-purpose solvers and other one-shot methods.

| EGN/SOTF-TopK/ours are one-shot solvers | k=30, m=500 | | | k=50, m=800 | | |
|---|---|---|---|---|---|---|
| | objective ↓ | optimal gap ↓ | time ↓ (sec) | objective ↓ | optimal gap ↓ | time ↓ (sec) |
| kmeans (Macqueen, 1967) | 3.020±0.200 | 0.214±0.049 | 0.053±0.074 | 2.905±0.159 | 0.233±0.042 | **0.085±0.164** |
| kmeans++ (Arthur & Vassilvitskii, 2007) | 2.854±0.166 | 0.169±0.043 | **0.042±0.007** | 2.693±0.101 | 0.174±0.027 | 0.504±0.234 |
| SCIP 7.0 (faster) | 4.641±1.880 | 0.377±0.288 | 71.738±13.192 | 5.450±0.674 | 0.587±0.046 | 218.426±57.530 |
| SCIP 7.0 (slower) | 4.470±1.918 | 0.348±0.295 | 118.068±48.055 | 5.258±1.018 | 0.552±0.146 | 243.919±54.118 |
| Gurobi 9.0 (faster) | 3.365±0.341 | 0.290±0.072 | 80.582±0.861 | 3.532±0.358 | 0.365±0.065 | 116.446±5.087 |
| Gurobi 9.0 (slower) | 2.453±0.142 | 0.033±0.042 | 125.589±0.606 | 3.364±0.268 | 0.335±0.055 | 214.360±3.785 |
| Gurobi 9.0 (optimal) | 2.365±0.063 | 0.000±0.000 | 314.798±116.858 | 2.221±0.041 | 0.000±0.000 | 648.213±194.486 |
| EGN (efficient) (Karalias & Loukas, 2020) | 3.032±0.195 | 0.217±0.048 | 0.830±0.308 | 2.865±0.138 | 0.223±0.036 | 0.988±0.140 |
| EGN (accurate) (Karalias & Loukas, 2020) | 2.795±0.140 | 0.152±0.035 | 123.559±12.278 | 2.815±0.124 | 0.209±0.034 | 191.091±13.141 |
| SOFT-TopK (Xie et al., 2020) | 2.719±0.136 | 0.129±0.037 | 97.147±6.565 | 2.600±0.104 | 0.145±0.029 | 120.835±7.298 |
| SOFT-TopK (w/ sampling) | 2.717±0.118 | 0.129±0.027 | 129.856±7.256 | 2.593±0.086 | 0.143±0.026 | 166.132±8.701 |
| GS-TopK (ours) | 2.420±0.072 | 0.023±0.009 | 76.534±6.321 | 2.283±0.050 | 0.027±0.008 | 120.689±2.405 |
| Homotopy GS-TopK (ours) | **2.418±0.076** | **0.022±0.010** | 103.742±4.778 | **2.273±0.047** | **0.023±0.007** | 158.400±3.498 |

## 4.2 EXPERIMENTAL RESULTS

Table 1 and Table 2 report results on max covering and discrete clustering problems, respectively. An interesting finding is that the classic algorithms are efficient but also very effective. However, if we can afford a higher time budget, there seems no universal approach to improve the results of the classic algorithms. In comparison, Gurobi and SCIP solvers can trade-off time for better accuracy, but they are less effective than our learning methods when considering the objective scores under the same time budget. Among one-shot learning methods, Erdos Goes Neural (Karalias & Loukas, 2020) is the only method that does not encode the constraint in the model output, and the experimental results suggest that it is appealing if we can encode the constraint in the model output. Directly applying Gumbel sampling to SOFT-TopK seems either an effective improvement, and satisfying results are achieved when exploiting the Gumbel re-parameterization trick and enabling gradient-based optimization over the combinatorial objective score. We further validate the effectiveness of Homotopy GS-TopK, where the objective scores are improved at the cost of more inference time.

**Further Discussions**. Our insight for the effectiveness of GS-TopK over other state-of-the-art one-shot learning methods (Karalias & Loukas, 2020; Xie et al., 2020) is that it can provide a more accurate estimation of the combinatorial objective score with discrete decision variables. Fig. 3 shows that our GS-TopK estimates a lower bound of the true objective score at the beginning 200 iterations while providing an accurate estimation afterward. Besides, the Homotopy version of our algorithm can gradually reduce the variance and help converge to a better solution.

REPRODUCIBILITY STATEMENT

The following efforts are made to ensure the reproducibility of this paper:

- We provide a complete proof about our Lemmas and Theorems in the Appendix, see Appendix A;
- We discuss the hyper-parameter configurations and the experiment setup in Sec 4.1;
- We provide the implementation details on max covering problem in Sec. 3.3, and the implementation details on discrete problem problem in Appendix B;
- We discuss the way of generating random dataset in Sec 4.1;
- The code will be made publicly available once this paper is accepted.

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

# A    PROOF OF THEOREMS

Before starting the detailed proof of the lemmas and theorems, firstly we recall the notations used in this paper:

- $\mathbf{T}^* = \mathtt{TopK}(\mathbf{D})$ is the optimal solution of the integer linear programming form of the OT problem Eq. (4), which is equivalent to the solution by firstly sorting all items and then select the TopK items;

- $\mathbf{T}^{\tau*} = \mathtt{Sinkhorn}(\mathbf{D})$ is the optimal solution of the entropic regularized form of the OT problem Eq. (5) solved by Sinkhorn algorithm. It is also the output by SOFT-TopK (Xie et al., 2020);

- $\mathbf{T}^{\sigma*} = \mathtt{TopK}(\widetilde{\mathbf{D}})$ is the optimal solution of the integer linear programming form of the OT problem after disturbed by the Gumbel noise Eq. (11), which is equivalent to the solution by firstly adding the Gumbel noise, then sorting all items and finally select the TopK items;

- $\widetilde{\mathbf{T}}^* = \mathtt{Sinkhorn}(\widetilde{\mathbf{D}})$ is the optimal solution of the entropic regularized form of the OT problem after disturbed by the Gumbel noise Eq. (13) solved by Sinkhorn algorithm. It is also the output by our proposed GS-TopK.

**Lemma 3.** *Given real numbers $x_i, x_j$, and $u_i, u_j$ are from i.i.d. $(0, 1)$ uniform distribution. After Gumbel perturbation, the probability that $x_i + g_\sigma(u_i) > x_j + g_\sigma(u_j)$ is:*

$$P(x_i + g_\sigma(u_i) > x_j + g_\sigma(u_j)) = \frac{1}{1 + \exp - \frac{x_i - x_j}{\sigma}}. \tag{21}$$

*Proof.* Since $g_\sigma(u_i) = -\sigma \log(-\log(u_i))$, $P(x_i + g_\sigma(u_i) > x_j + g_\sigma(u_j))$ is equivalent to the probability that the following inequality holds:

$$x_i - \sigma \log(-\log(u_i)) > x_j - \sigma \log(-\log(u_j)) \tag{22}$$

And we have

$$x_i - x_j > \sigma \log(-\log(u_i)) - \sigma \log(-\log(u_j)) \tag{23}$$

$$\frac{x_i - x_j}{\sigma} > \log\left(\frac{\log(u_i)}{\log(u_j)}\right) \tag{24}$$

$$e^{\frac{x_i - x_j}{\sigma}} > \frac{\log(u_i)}{\log(u_j)} \tag{25}$$

Since $u_j \in (0, 1)$, $\log(u_j) < 0$. Then we have

$$\log(u_j) < \log(u_i) e^{-\frac{x_i - x_j}{\sigma}} \tag{26}$$

$$\log(u_j) < \log\left(u_i^{\exp - \frac{x_i - x_j}{\sigma}}\right) \tag{27}$$

$$u_j < u_i^{\exp - \frac{x_i - x_j}{\sigma}} \tag{28}$$

Since $u_i, u_j$ are i.i.d. uniform distributions, the probability when the above formula holds is

$$\int_0^1 \int_0^{u_i^{\exp - \frac{x_i - x_j}{\sigma}}} du_j \, du_i = \int_0^1 u_i^{\exp - \frac{x_i - x_j}{\sigma}} du_i = \frac{1}{1 + \exp - \frac{x_i - x_j}{\sigma}} \tag{29}$$

Thus the probability that $x_i + g_\sigma(u_i) > x_j + g_\sigma(u_j)$ after Gumbel perturbation is:

$$P(x_i + g_\sigma(u_i) > x_j + g_\sigma(u_j)) = \frac{1}{1 + \exp - \frac{x_i - x_j}{\sigma}} \tag{30}$$

$\square$

### A.1 PROOF OF LEMMA 1

*Proof of Lemma 1.* If one of the top-$k$ items becomes smaller than $x_{k+1}$ after distorted by the Gumbel noise, then $||\mathbf{T}^* - \mathbf{T}^{\sigma*}||_F$ increases at most by 2. Without loss of generality, here we assume $x_i$ as one of the top-$k$ items i.e. $i \le k$. According to **Lemma 3**, the probability that $x_i$ becomes smaller than $x_{k+1}$ after perturbation is

$$P(x_i + g_\sigma(u_i) < x_{k+1} + g_\sigma(u_{k+1})) = \frac{1}{1 + \exp - \frac{x_{k+1} - x_i}{\sigma}} \tag{31}$$

Similarly, if one of the last-$(m - k)$ items becomes larger than $x_k$ after distorted by the Gumbel noise, then $||\mathbf{T}^* - \mathbf{T}^{\sigma*}||_F$ also increases at most by 2. Without loss of generality, here we assume $x_j$ as one of the last-$(m - k)$ items i.e. $j \ge k + 1$. According to **Lemma 3**, the probability that $x_j$ becomes larger than $x_k$ after perturbation is

$$P(x_j + g_\sigma(u_j) > x_k + g_\sigma(u_k)) = \frac{1}{1 + \exp - \frac{x_j - x_k}{\sigma}} \tag{32}$$

To compute the expectation of $||\mathbf{T}^* - \mathbf{T}^{\sigma*}||$, we sum all probabilities and we have

$$\mathbb{E}_u \left[ ||\mathbf{T}^* - \mathbf{T}^{\sigma*}||_F^2 \right] \le \sum_{i=1}^{k} \frac{4}{1 + \exp - \frac{x_{k+1} - x_i}{\sigma}} + \sum_{j=k+1}^{m} \frac{4}{1 + \exp - \frac{x_j - x_k}{\sigma}} \tag{33}$$

where the constant 4 appears because $||\mathbf{T}^* - \mathbf{T}^{\sigma*}||_F^2$ will change at most by 4 if one more item crosses the boundary. Since for all $i \le k$ and $j \ge k + 1$, we have

$$x_{k+1} - x_i \le x_{k+1} - x_k \tag{34}$$
$$x_j - x_k \le x_{k+1} - x_k \tag{35}$$

Thus we have

$$\text{Eq. (33)} \le \sum_{i=1}^{k} \frac{4}{1 + \exp - \frac{x_{k+1} - x_k}{\sigma}} + \sum_{j=k+1}^{m} \frac{4}{1 + \exp - \frac{x_{k+1} - x_k}{\sigma}} \tag{36}$$

$$= \sum_{i=1}^{m} \frac{4}{1 + \exp - \frac{x_{k+1} - x_k}{\sigma}} \tag{37}$$

$$= \frac{4m}{1 + \exp \frac{|x_k - x_{k+1}|}{\sigma}} \tag{38}$$

Since $\mathbb{E}_u \left[ ||\mathbf{T}^* - \mathbf{T}^{\sigma*}||_F \right] \le \sqrt{\mathbb{E}_u \left[ ||\mathbf{T}^* - \mathbf{T}^{\sigma*}||_F^2 \right]}$, we have

$$\mathbb{E}_u \left[ ||\mathbf{T}^* - \mathbf{T}^{\sigma*}||_F \right] \le \sqrt{\frac{4m}{1 + \exp \frac{|x_k - x_{k+1}|}{\sigma}}} \tag{39}$$

$\square$

### A.2 PROOF OF LEMMA 2

*Proof of Lemma 2.* By disturbing $\mathcal{X}$ with i.i.d. Gumbel noise, we have

$$\mathcal{X}' = x_1 + g_\sigma(u_1), x_2 + g_\sigma(u_2), x_3 + g_\sigma(u_3), ..., x_m + g_\sigma(u_m) \tag{40}$$

where $g_\sigma(u) = -\sigma \log(-\log(u))$ is the Gumbel noise modulated by noise factor $\sigma$, and $u_1, u_2, u_3, ..., u_m$ are i.i.d. uniform distribution. We define $\pi$ as the permutation of sorting $\mathcal{X}'$ in descending order, i.e. $x_{\pi_1} + g_\sigma(u_{\pi_1}), x_{\pi_2} + g_\sigma(u_{\pi_2}), x_{\pi_3} + g_\sigma(u_{\pi_3}), ..., x_{\pi_m} + g_\sigma(u_{\pi_m})$ are in descending order.

Recall **Theorem 2 in** ([Xie et al., 2020](#)), for $x_1, x_2, x_3, ..., x_m$ we have

$$||\mathbf{T}^* - \mathbf{T}^{\tau*}||_F \leq \frac{2m\tau \log 2}{|x_k - x_{k+1}|} \tag{41}$$

By substituting $\mathcal{X}$ with $\mathcal{X}'$ and taking the expected value, we have

$$\mathbb{E}_u \left[ ||\mathbf{T}^{\sigma*} - \widetilde{\mathbf{T}}^*||_F \right] \leq \mathbb{E}_u \left[ \frac{2m\tau \log 2}{|x_{\pi_k} + g_\sigma(u_{\pi_k}) - x_{\pi_{k+1}} - g_\sigma(u_{\pi_{k+1}})|} \right] \tag{42}$$

Based on **Lemma 3**, the probability that $\pi_k = i, \pi_{k+1} = j$ is

$$P(\pi_k = i, \pi_{k+1} = j) = \frac{1}{1 + \exp -\frac{x_i - x_j}{\sigma}} \sum_{\forall \pi} \left( \prod_{a=1}^{k-1} \frac{1}{1 + \exp -\frac{x_{\pi_a} - x_i}{\sigma}} \prod_{b=k+2}^{m} \frac{1}{1 + \exp -\frac{x_j - x_{\pi_b}}{\sigma}} \right) \tag{43}$$

where the first term denotes $x_i + g_\sigma(u_i) > x_j + g_\sigma(u_j)$, the second term denotes all conditions that there are $(k-1)$ items larger than $x_i + g_\sigma(u_i)$ and the rest items are smaller than $x_j + g_\sigma(u_j)$.

In the following we derive the upper bound of $\mathbb{E}_u \left[ \frac{1}{|x_{\pi_k} + g_\sigma(u_{\pi_k}) - x_{\pi_{k+1}} - g_\sigma(u_{\pi_{k+1}})|} \right]$. We denote $\mathcal{A}_{i,j}$ as

$$u_i, u_j \in \mathcal{A}_{i,j}, \quad s.t. \quad x_i + g_\sigma(u_i) - x_j - g_\sigma(u_j) > \epsilon \tag{44}$$

where $\epsilon$ is a sufficiently small number. Then we have

$$\mathbb{E}_u \left[ \frac{1}{|x_{\pi_k} + g_\sigma(u_{\pi_k}) - x_{\pi_{k+1}} - g_\sigma(u_{\pi_{k+1}})|} \right]$$

$$= \sum_{i \neq j} P(\pi_k = i, \pi_{k+1} = j) \, \mathbb{E}_{u_i, u_j \in \mathcal{A}_{i,j}} \left[ \frac{1}{|x_i + g_\sigma(u_i) - x_j - g_\sigma(u_j)|} \right] \tag{45}$$

$$= \sum_{i \neq j} \left( \frac{1}{1 + \exp -\frac{x_i - x_j}{\sigma}} \sum_{\forall \pi} \left( \prod_{a=1}^{k-1} \frac{1}{1 + \exp -\frac{x_{\pi_a} - x_i}{\sigma}} \prod_{b=k+2}^{m} \frac{1}{1 + \exp -\frac{x_j - x_{\pi_b}}{\sigma}} \right) \right.$$
$$\left. \mathbb{E}_{u_i, u_j \in \mathcal{A}_{i,j}} \left[ \frac{1}{|x_i + g_\sigma(u_i) - x_j - g_\sigma(u_j)|} \right] \right) \tag{46}$$

$$= \sum_{i \neq j} \left( \frac{1}{1 + \exp -\frac{x_i - x_j}{\sigma}} \sum_{\forall \pi} \left( \prod_{a=1}^{k-1} \frac{1}{1 + \exp -\frac{x_{\pi_a} - x_i}{\sigma}} \prod_{b=k+2}^{m} \frac{1}{1 + \exp -\frac{x_j - x_{\pi_b}}{\sigma}} \right) \right.$$
$$\left. \mathbb{E}_{u_i, u_j \in \mathcal{A}_{i,j}} \left[ \frac{1}{|x_i - \sigma \log(-\log(u_i)) - x_j + \sigma \log(-\log(u_j))|} \right] \right) \tag{47}$$

$$= \sum_{i \neq j} \left( f(x_i - x_j, \sigma, z) \sum_{\forall \pi} \left( \prod_{a=1}^{k-1} \frac{1}{1 + \exp -\frac{x_{\pi_a} - x_i}{\sigma}} \prod_{b=k+2}^{m} \frac{1}{1 + \exp -\frac{x_j - x_{\pi_b}}{\sigma}} \right) \right) \tag{48}$$

We denote $f(\delta, \sigma, z)$ as:

$$f(\delta, \sigma, z) = \frac{1}{1 + \exp -\frac{\delta}{\sigma}} \mathbb{E}_{u_i, u_j} \left[ \frac{1}{|\delta - \sigma \log(-\log(u_i)) + \sigma \log(-\log(u_j))|} \right] \tag{49}$$

$$s.t. \quad \delta - \sigma \log(-\log(u_i)) + \sigma \log(-\log(u_j)) > z > 0 \tag{50}$$

For the probability terms in Eq. ([48](#)), for all permutations $\pi$, there must exist $\pi_a, \pi_b$, such that

$$\frac{1}{1 + \exp -\frac{x_{\pi_a} - x_i}{\sigma}} \leq \frac{1}{1 + \exp -\frac{x_k - x_i}{\sigma}} \tag{51}$$

$$\frac{1}{1 + \exp -\frac{x_j - x_{\pi_b}}{\sigma}} \leq \frac{1}{1 + \exp -\frac{x_j - x_{k+1}}{\sigma}} \tag{52}$$

Thus we have

$$\text{Eq. (48)} \leq \sum_{i \neq j} \left( f(x_i - x_j, \sigma, z) \frac{1}{1 + \exp{-\frac{x_k - x_i}{\sigma}}} \frac{1}{1 + \exp{-\frac{x_j - x_{k+1}}{\sigma}}} \right) \tag{53}$$

$$\leq \sum_{i \neq j} \frac{f(x_i - x_j, \sigma, z)}{(1 + \exp{\frac{x_i - x_k}{\sigma}})(1 + \exp{\frac{x_{k+1} - x_j}{\sigma}})} \tag{54}$$

By transforming Eq. (21) in **Lemma 3**, and substituting $x_j - x_i$ by $y$, we have

$$\text{Eq. (21)} \Rightarrow P(g_\sigma(u_i) - g_\sigma(u_j) > x_j - x_i) = \frac{1}{1 + \exp{-\frac{x_i - x_j}{\sigma}}} \tag{55}$$

$$\Rightarrow P(g_\sigma(u_i) - g_\sigma(u_j) > y) = \frac{1}{1 + \exp{\frac{y}{\sigma}}} \tag{56}$$

$$\Rightarrow P(g_\sigma(u_i) - g_\sigma(u_j) < y) = 1 - \frac{1}{1 + \exp{\frac{y}{\sigma}}} = \frac{1}{1 + \exp{-\frac{y}{\sigma}}} \tag{57}$$

where the right side is the form of the cumulative distribution function (CDF) of standard Logistic distribution by setting $\sigma = 1$:

$$\text{CDF}(y) = \frac{1}{1 + \exp{(-y)}} \tag{58}$$

Thus $-\log(-\log(u_i)) + \log(-\log(u_j))$ is equivalent to the Logistic distribution whose probability density function (PDF) is

$$\text{PDF}(y) = \frac{d\text{CDF}(y)}{dy} = \frac{1}{\exp{(-y)} + \exp{y} + 2} \tag{59}$$

and in this proof we exploit an upper bound of $\text{PDF}(y)$:

$$\text{PDF}(y) = \frac{1}{\exp{(-y)} + \exp{y} + 2} \leq \frac{1}{y^2 + 4} \tag{60}$$

Based on the Logistic distribution, we can replace $-\sigma \log(-\log(u_i)) + \sigma \log(-\log(u_j))$ by $\sigma y$ where $y$ is from the Logistic distribution. Thus we can derive the upper bound of $f(\delta, \sigma, z)$ as

follows

$$f(\delta, \sigma, z) = \frac{1}{1 + \exp{-\frac{\delta}{\sigma}}} \cdot \frac{\int_{-\delta/\sigma + z}^{\infty} \frac{1}{\delta + \sigma y} \frac{1}{\exp(-y) + \exp y + 2} dy}{\int_{-\delta/\sigma + z}^{\infty} \frac{1}{\exp(-y) + \exp y + 2} dy} \tag{61}$$

$$= \frac{1}{1 + \exp{-\frac{\delta}{\sigma}}} \cdot \frac{\int_{-\delta/\sigma + z}^{\infty} \frac{1}{\delta + \sigma y} \frac{1}{\exp(-y) + \exp y + 2} dy}{1 - \frac{1}{1 + \exp(\delta/\sigma - z)}} \tag{62}$$

$$= \frac{1}{1 + \exp{-\frac{\delta}{\sigma}}} \cdot \frac{\int_{-\delta/\sigma + z}^{\infty} \frac{1}{\delta + \sigma y} \frac{1}{\exp(-y) + \exp y + 2} dy}{\frac{\exp(\delta/\sigma - z)}{1 + \exp(\delta/\sigma - z)}} \tag{63}$$

$$= \frac{1}{1 + \exp{-\frac{\delta}{\sigma}}} \cdot \frac{\int_{-\delta/\sigma + z}^{\infty} \frac{1}{\delta + \sigma y} \frac{1}{\exp(-y) + \exp y + 2} dy}{\frac{1}{1 + \exp(-\delta/\sigma + z)}} \tag{64}$$

$$= \frac{1 + \exp(-\frac{\delta}{\sigma} + z)}{1 + \exp{-\frac{\delta}{\sigma}}} \int_{-\delta/\sigma + z}^{\infty} \frac{1}{\delta + \sigma y} \frac{1}{\exp(-y) + \exp y + 2} dy \tag{65}$$

$$\leq \frac{1 + \exp(-\frac{\delta}{\sigma} + z)}{1 + \exp{-\frac{\delta}{\sigma}}} \int_{-\delta/\sigma + z}^{\infty} \frac{1}{\delta + \sigma y} \frac{1}{y^2 + 4} dy \tag{66}$$

$$= \frac{1 + \exp(-\frac{\delta}{\sigma} + z)}{1 + \exp{-\frac{\delta}{\sigma}}} \cdot \frac{2\sigma \log\left((z\sigma - \delta)^2 + 4\sigma^2\right) - 2\delta \arctan\left(\frac{z - \delta/\sigma}{2}\right) - 4\sigma \log z + \pi\delta}{4\delta^2 + 16\sigma^2} \tag{67}$$

$$\leq \frac{1 + \exp(-\frac{\delta}{\sigma} + z)}{1 + \exp{-\frac{\delta}{\sigma}}} \cdot \frac{2\sigma \log\left((z\sigma + |\delta|)^2 + 4\sigma^2\right) - 2\delta \arctan\left(\frac{z - \delta/\sigma}{2}\right) - 4\sigma \log z + \pi\delta}{4\delta^2 + 16\sigma^2} \tag{68}$$

$$= \frac{1 + \exp(-\frac{\delta}{\sigma} + z)}{1 + \exp{-\frac{\delta}{\sigma}}} \cdot \frac{2\sigma \log\left((z\sigma + |\delta|)^2 + 4\sigma^2\right) - 2\delta \arctan\left(\frac{z - \delta/\sigma}{2}\right) - 2\sigma \log z^2 + \pi\delta}{4\delta^2 + 16\sigma^2} \tag{69}$$

$$= \frac{1 + \exp(-\frac{\delta}{\sigma} + z)}{1 + \exp{-\frac{\delta}{\sigma}}} \cdot \frac{2\sigma \log\left(\frac{(z\sigma + |\delta|)^2 + 4\sigma^2}{z^2}\right) - 2\delta \arctan\left(\frac{z - \delta/\sigma}{2}\right) + \pi\delta}{4\delta^2 + 16\sigma^2} \tag{70}$$

$$\leq \frac{1 + \exp(-\frac{\delta}{\sigma} + z)}{1 + \exp{-\frac{\delta}{\sigma}}} \cdot \frac{2\sigma \log\left(\frac{(z\sigma + |\delta| + 2\sigma)^2}{z^2}\right) - 2\delta \arctan\left(\frac{z - \delta/\sigma}{2}\right) + \pi\delta}{4\delta^2 + 16\sigma^2} \tag{71}$$

$$= \frac{1 + \exp(-\frac{\delta}{\sigma} + z)}{1 + \exp{-\frac{\delta}{\sigma}}} \cdot \frac{4\sigma \log\left(\frac{z\sigma + |\delta| + 2\sigma}{z}\right) - 2\delta \arctan\left(\frac{z - \delta/\sigma}{2}\right) + \pi\delta}{4\delta^2 + 16\sigma^2} \tag{72}$$

$$= \frac{1 + \exp(-\frac{\delta}{\sigma} + z)}{1 + \exp{-\frac{\delta}{\sigma}}} \cdot \frac{4\sigma \log\left(\frac{z\sigma + |\delta| + 2\sigma}{z}\right) + \delta\left(\pi - 2\arctan\left(\frac{z - \delta/\sigma}{2}\right)\right)}{4\delta^2 + 16\sigma^2} \tag{73}$$

$$\leq \frac{1 + \exp(-\frac{\delta}{\sigma} + z)}{1 + \exp{-\frac{\delta}{\sigma}}} \cdot \frac{4\sigma \log\left(\frac{z\sigma + |\delta| + 2\sigma}{z}\right) + |\delta|\left(\pi - 2\arctan\left(\frac{z - \delta/\sigma}{2}\right)\right)}{4\delta^2 + 16\sigma^2} \tag{74}$$

$$\leq \frac{1 + \exp(-\frac{\delta}{\sigma} + z)}{1 + \exp{-\frac{\delta}{\sigma}}} \cdot \frac{4\sigma \log\left(\frac{z\sigma + |\delta| + 2\sigma}{z}\right) + |\delta|\left(\pi - 2\arctan\left(-\frac{\delta}{2\sigma}\right)\right)}{4\delta^2 + 16\sigma^2} \tag{75}$$

$$= \frac{1 + \exp(-\frac{\delta}{\sigma} + z)}{1 + \exp{-\frac{\delta}{\sigma}}} \cdot \frac{4\sigma \log\left(\frac{z\sigma + |\delta| + 2\sigma}{z}\right) + |\delta|\left(\pi + 2\arctan\left(\frac{\delta}{2\sigma}\right)\right)}{4\delta^2 + 16\sigma^2} \tag{76}$$

where Eq. (66) is because $\frac{1}{\exp(-y)+\exp y+2} \leq \frac{1}{y^2+4}$, and Eq. (74) is because $\pi - 2\arctan(\frac{z-\delta/\sigma}{2}) \geq 0$. With probability $(1-\epsilon)$, we have

$$z = \log \frac{1+\epsilon \exp \frac{\delta}{\sigma}}{1-\epsilon} \geq -\log(1-\epsilon) \tag{77}$$

$$\frac{1+\exp\left(-\frac{\delta}{\sigma}+z\right)}{1+\exp -\frac{\delta}{\sigma}} = \frac{1}{1-\epsilon} \tag{78}$$

Thus

$$f(\delta,\sigma,z) \leq \text{Eq. (76)} = \frac{1}{1-\epsilon} \frac{4\sigma \log\left(\frac{z\sigma+|\delta|+2\sigma}{z}\right) + |\delta|\left(\pi + 2\arctan\left(\frac{\delta}{2\sigma}\right)\right)}{4\delta^2 + 16\sigma^2} \tag{79}$$

$$\leq \frac{1}{1-\epsilon} \frac{4\sigma \log\left(\sigma - \frac{|\delta|+2\sigma}{\log(1-\epsilon)}\right) + |\delta|\left(\pi + 2\arctan\left(\frac{\delta}{2\sigma}\right)\right)}{4\delta^2 + 16\sigma^2} \tag{80}$$

Thus we have

$$\text{Eq. (54)} \leq \sum_{i\neq j} \left( \frac{4\sigma \log\left(\sigma - \frac{|x_i-x_j|+2\sigma}{\log(1-\epsilon)}\right) + |x_i-x_j|\left(\pi + 2\arctan\left(\frac{x_i-x_j}{2\sigma}\right)\right)}{(1-\epsilon)(4(x_i-x_j)^2 + 16\sigma^2)(1+\exp\frac{x_i-x_k}{\sigma})(1+\exp\frac{x_{k+1}-x_j}{\sigma})} \right) \tag{81}$$

In conclusion, with the probability $(1-\epsilon)$, we have

$$\mathbb{E}_u\left[||\mathbf{T}^{\sigma*} - \widetilde{\mathbf{T}}^*||_F\right] \leq \sum_{i\neq j} \frac{(2\log 2)m\tau \left(4\sigma \log\left(\sigma - \frac{|x_i-x_j|+2\sigma}{\log(1-\epsilon)}\right) + |x_i-x_j|\left(\pi + 2\arctan\frac{x_i-x_j}{2\sigma}\right)\right)}{(1-\epsilon)(4(x_i-x_j)^2 + 16\sigma^2)(1+\exp\frac{x_i-x_k}{\sigma})(1+\exp\frac{x_{k+1}-x_j}{\sigma})} \tag{82}$$

$$= \sum_{i\neq j} \frac{(\log 2)m\tau \left(2\sigma \log\left(\sigma - \frac{|x_i-x_j|+2\sigma}{\log(1-\epsilon)}\right) + |x_i-x_j|\left(\frac{\pi}{2} + \arctan\frac{x_i-x_j}{2\sigma}\right)\right)}{(1-\epsilon)((x_i-x_j)^2 + 4\sigma^2)(1+\exp\frac{x_i-x_k}{\sigma})(1+\exp\frac{x_{k+1}-x_j}{\sigma})} \tag{83}$$

$$= (\log 2)m\tau \sum_{i\neq j} \Omega(x_i,x_j,\sigma,\epsilon) \tag{84}$$

And we denote $\Omega(x_i,x_j,\sigma,\epsilon)$ as

$$\Omega(x_i,x_j,\sigma,\epsilon) = \frac{2\sigma \log\left(\sigma - \frac{|x_i-x_j|+2\sigma}{\log(1-\epsilon)}\right) + |x_i-x_j|\left(\frac{\pi}{2} + \arctan\frac{x_i-x_j}{2\sigma}\right)}{(1-\epsilon)((x_i-x_j)^2 + 4\sigma^2)(1+\exp\frac{x_i-x_k}{\sigma})(1+\exp\frac{x_{k+1}-x_j}{\sigma})} \tag{85}$$

$\square$

Finally, **Theorem 1** is proved by triangle inequality by jointly considering **Lemma 1** and **Lemma 2**.

## A.3 REMARKS W.R.T. SOFT-TOPK

In the following, we add some remarks about the relationship between our conclusion of GS-TopK and the conclusion derived by the authors of SOFT-TopK (Xie et al., 2020): the SOFT-TopK is a special case of our proposed algorithm when we set $\sigma = 0$. We have the following conclusion:

**Theorem 2.** *Assume the values of $x_k, x_{k+1}$ are unique[4], under probability $(1-\epsilon)$, we have*

$$\lim_{\sigma\to 0^+} \mathbb{E}_u\left[||\mathbf{T}^* - \widetilde{\mathbf{T}}^*||_F\right] \leq \frac{(\pi\log 2)m\tau}{(1-\epsilon)|x_k - x_{k+1}|} \tag{86}$$

---

[4]For the ease of a compact proof, we make this assumption that the values of $x_k, x_{k+1}$ are unique. If there are duplicate values of $x_k, x_{k+1}$, the bound only differs by a constant multiplier therefore does not affect our conclusion: SOFT-TopK (Xie et al., 2020) is a special case of our proposed approach when $\sigma = 0$.

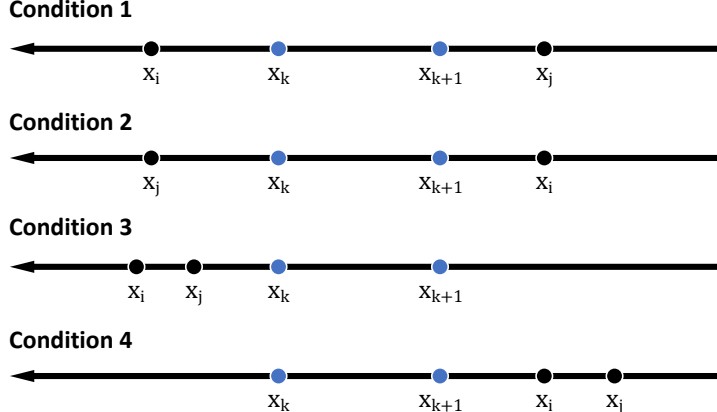

Figure 4: Four conditions considered in our proof. It is worth noting that $x_i, x_j$ must not lie between $x_k, x_{k+1}$, because we define $x_k, x_{k+1}$ as two adjacent items in the original sorted list.

which differs from the conclusion in (Xie et al., 2020) by only a constant factor.

*Proof.* Since $\sigma \to 0+$, the first term becomes 0. For the second term, we discuss four conditions as shown in Fig. 4, except for the following condition: $x_i = x_k, x_j = x_{k+1}$.

**Condition 1**. If $x_i \geq x_k, x_j \leq x_{k+1}$ (equalities do not hold at the same time), we have at least $x_i - x_k > 0$ or $x_{k+1} - x_j > 0$. Then we have

$$\lim_{\sigma \to 0^+} \frac{1}{(1 + \exp \frac{x_i - x_k}{\sigma})(1 + \exp \frac{x_{k+1} - x_j}{\sigma})} = 0 \tag{87}$$

$$\Rightarrow \lim_{\sigma \to 0^+} \Omega(x_i, x_j, \sigma, \epsilon) = 0 \tag{88}$$

**Condition 2**. For any case that $x_i < x_j$, we have $x_i - x_j < 0$, thus

$$\lim_{\sigma \to 0^+} \arctan \frac{x_i - x_j}{\sigma} = -\frac{\pi}{2} \tag{89}$$

$$\Rightarrow \lim_{\sigma \to 0^+} \frac{\pi}{2} + \arctan \frac{x_i - x_j}{\sigma} = 0 \tag{90}$$

$$\Rightarrow \lim_{\sigma \to 0^+} \Omega(x_i, x_j, \sigma, \epsilon) = 0 \tag{91}$$

**Condition 3**. If $x_i \geq x_j \geq x_k$ (equalities do not hold at the same time), we have $x_i - x_k > 0$. Then we have

$$\lim_{\sigma \to 0^+} \frac{1}{1 + \exp \frac{x_i - x_k}{\sigma}} = 0 \tag{92}$$

$$\Rightarrow \lim_{\sigma \to 0^+} \Omega(x_i, x_j, \sigma, \epsilon) = 0 \tag{93}$$

**Condition 4**. If $x_{k+1} \geq x_i \geq x_j$ (equalities do not hold at the same time), we have $x_{k+1} - x_j > 0$. Then we have

$$\lim_{\sigma \to 0^+} \frac{1}{1 + \exp \frac{x_{k+1} - x_j}{\sigma}} = 0 \tag{94}$$

$$\Rightarrow \lim_{\sigma \to 0^+} \Omega(x_i, x_j, \sigma, \epsilon) = 0 \tag{95}$$

Therefore, if $x_i \neq x_k$ and $x_j \neq x_{k+1}$, the second term $\Omega(x_i, x_j, \sigma, \epsilon)$ degenerates to 0 when $\sigma \to 0^+$. Thus we have the following conclusion by only considering $x_i = x_k, x_j = x_{k+1}$:

$$\lim_{\sigma \to 0^+} \mathbb{E}_u \left[ ||\mathbf{T}^* - \widetilde{\mathbf{T}}^*||_F \right] \leq \frac{(\log 2) m \tau \left( |x_k - x_{k+1}| \left( \frac{\pi}{2} + \arctan \frac{x_k - x_{k+1}}{2\sigma} \right) \right)}{(1-\epsilon)(x_k - x_{k+1})^2} \tag{96}$$

$$\leq \frac{(\pi \log 2) m \tau}{(1-\epsilon)|x_k - x_{k+1}|} \tag{97}$$

$\square$

## B    IMPLEMENTATION DETAILS ON DISCRETE CLUSTERING PROBLEM

For the discrete clustering problem, recall that we aim to select a subset of objects as cluster centers. It differs from the k-means clustering problem whose cluster centers may not have to be a subset of the objects, and the constraint for discrete clustering is more welcomed for real-world facility location planning applications. Denote $\mathbf{\Delta} \in \mathbb{R}^{+ m \times m}$ as the distance matrix for each pair of objects.

Following our implementation of the max covering problem, we can also derive a GPU-friendly formulation of the discrete clustering problem using PyTorch syntax:

$$\min_{\mathbf{x}} \text{sum}(\min(\mathbf{\Delta}[\mathbf{x}, :], dim = 0)) \qquad s.t. \quad \mathbf{x} \in \{0, 1\}^m, ||x||_0 \leq k \tag{98}$$

We also model the discrete clustering problem as a graph and encode the problem structure by a graph neural network. Specifically, for points that lie on the 2D plane, we define there is an edge between two points if their distance is smaller than a predefined threshold. Then the geometric information is encoded by 3-layer SplineCNN (Fey et al., 2018) which was found successful for geometric feature learning. Similar to the max covering problem, we rewrite the GPU-friendly form of the objective score in a differentiable manner, being estimated by the output of GS-TopK algorithm:

$$\widetilde{J}_i = \text{sum}(\text{softmax}(-T\mathbf{\Delta}[\mathbf{x}, :], dim = 0, keepdim = True) \odot \mathbf{\Delta}[\mathbf{x}, :])$$
$$J = \text{mean}([\widetilde{J}_1, \widetilde{J}_2, ..., \widetilde{J}_{\#\text{G}}]) \tag{99}$$

where we use a softmax operator with temperature $T$ to approximate the $\min$ operator in a differentiable way. $\odot$ means the tensor-product operator in PyTorch. The GS-TopK learning algorithm is summarized in Alg. 3.

## C    EXPERIMENT SETUP

Our algorithms are implemented by PyTorch and the graph neural network modules are based on (Fey & Lenssen, 2019). In our paper, we optimize the hyperparameters by greedy search on a small subset of problem instances ($\sim$5) and set the best configuration of hyperparameters for both GS-TopK, SOFT-TopK. For the max covering problem, we empirically set learning rate $= 0.1, \tau = 0.05, \sigma = 0.15$ for GS-TopK, $\tau = 0.05$ for SOFT-TopK, and $(\tau, \sigma) = (0.05, 0.15), (0.04, 0.10), (0.03, 0.05)$ for Homotopy GS-TopK. For the discrete clustering problem, we set learning rate $= 0.1, \tau = 0.05, \sigma = 0.25$ for GS-TopK, $\tau = 0.05$ for SOFT-TopK, and we set $(\tau, \sigma) = (0.05, 0.25), (0.04, 0.15), (0.03, 0.05)$ for Homotopy GS-TopK. We set $\#G = 1000$ for max covering, $\#G = 500$ for discrete clustering. All experiments are done on a workstation with i7-9700K@3.60GHz, 16GB memory, and 2080Ti GPU.

## D    ABLATION STUDY ON HYPERPARAMETERS

Firstly, we want to add some remarks about the selection of hyperparameters:

- $\#G$ **(number of Gumbel samples):** $\#G$ affects how many samples are taken during training and inference for GS-TopK. A larger $\#G$ (i.e. more samples) will be more appealing,

---

**Algorithm 3: Gumbel-Sinkhorn-TopK Learning for Solving the Discrete Clustering Problem**

---

**Input:** the distance matrix $\boldsymbol{\Delta}$; learning rate $\alpha$; softmax temperature $T$; GS-TopK parameters $k, \tau, \sigma, \#\mathrm{G}$.

1 **if** *Training* **then**
2    Randomly initialize neural network weights $\theta$;
3 **if** *Inference* **then**
4    Load a pretrained neural network weights $\theta$; $J_{best} = +\infty$;
5 **while** *not converged* **do**
6    $\mathbf{s} = \mathrm{SplineCNN}_\theta(\boldsymbol{\Delta})$; $[\widetilde{\mathbf{T}}_1, \widetilde{\mathbf{T}}_2, ..., \widetilde{\mathbf{T}}_{\#\mathrm{G}}] = \mathrm{GS\text{-}TopK}(\mathbf{s}, k, \tau, \sigma, \#\mathrm{G})$;
7    for all $i$,
     $\widetilde{J}_i = \mathrm{sum}(\mathrm{softmax}(-T\boldsymbol{\Delta}[\widetilde{\mathbf{T}}_i[2,:],:], dim = 0, keepdim = True) \odot \boldsymbol{\Delta}[\widetilde{\mathbf{T}}_i[2,:],:])$;
8    $J = \mathrm{mean}([\widetilde{J}_1, \widetilde{J}_2, ..., \widetilde{J}_{\#\mathrm{G}}])$;
9    **if** *Training* **then**
10      update $\theta$ with respect to the gradient $\frac{\partial J}{\partial \theta}$ and learning rate $\alpha$ by gradient descend;
11    **if** *Inference* **then**
12      update $\mathbf{s}$ with respect to the gradient $\frac{\partial J}{\partial \mathbf{s}}$ and learning rate $\alpha$ by gradient descend;
13      for all $i$, $\widetilde{J}_i = \mathrm{TopK}(\widetilde{\mathbf{T}}_i[2,:])\mathbf{A} \cdot \mathbf{v}$; $J_{best} = \min([\widetilde{J}_1, \widetilde{J}_2, ..., \widetilde{J}_{\#\mathrm{G}}], J_{best})$;
14 **if** *Homotopy Inference* **then**
15    Decline the values of $\tau, \sigma$ and jump to line 5;

**Output:** Learned network weights $\theta$ (if training)/The best objective $J_{best}$ (if inference).

---

because GS-TopK will have a more accurate estimation of the objective score, and it will have a higher probability of discovering better solutions. However, $\#G$ cannot be arbitrarily large because we are running GS-TopK in parallel on GPU, and the GPU has limited memory. Therefore, in experiments, we set a large enough $\#G$ (e.g. $\#G = 1000$) and ensure that it can fit into the GPU memory of our workstation (2080Ti, 11G).

- $\tau$ **(entropic regularization factor of Sinkhorn):** Theoretically, $\tau$ controls the gap of the continuous Sinkhorn solution to the discrete solution, and a smaller $\tau$ will lead to a tightened gap. This property is validated by our theoretical findings in **Lemma 2**. Unfortunately, $\tau$ cannot be arbitrarily small, because a smaller $\tau$ requires more Sinkhorn iterations to converge. Therefore, given a fixed number of Sinkhorn iterations (100) to ensure the efficiency of our algorithm, we need trial-and-error to discover the suitable $\tau$ for both SOFT-TopK and GS-TopK. The grid search results below show that our selection of $\tau$ fairly balances the performances of both SOFT-TopK and GS-TopK.

- $\sigma$ **(Gumbel noise factor):** As derived in **Theorem 1**, we need to balance between the two gaps (**Lemma 1** and **Lemma 2**) by a suitable $\sigma$. Since $|x_k - x_{k+1}|$ is unknown, we cannot directly find the theoretically optimal $\sigma$ given $\tau$. In the experiments, we firstly determine a $\tau$, and then find a suitable $\sigma$ by greedy search on a small subset ($\sim$5) of problem instances.

We conduct an ablation study about the sensitivity of hyperparameters by performing an extensive grid search near the configuration used in our max covering experiments ($\tau = 0.05, \sigma = 0.15, \#G = 1000$). We choose the k=50, m=500, n=1000 max covering problem, and we have the following results for GS-TopK (ours) and SOFT-TopK (Xie et al., 2020) (higher is better):

Table 3: Ablation study result of GS-TopK with $\#G = 1000$.

| $\sigma = $    $\tau = $ | 0.01 | 0.05 | 0.1 |
|---|---|---|---|
| 0.1 | 42513.4 | 44759.2 | **45039.5** |
| 0.15 | 41456.5 | 44713.2 | 44837.2 |
| 0.2 | 41264.3 | 44638.1 | 44748.2 |

Table 4: Ablation study result of GS-TopK with $\#G = 800$.

| $\sigma = $ \ $\tau = $ | 0.01 | 0.05 | 0.1 |
|---|---|---|---|
| 0.1 | 42511.6 | 44754.6 | **45037.6** |
| 0.15 | 41421.4 | 44705.8 | 44841.5 |
| 0.2 | 41235.9 | 44651.5 | 44748.6 |

Table 5: Ablation study result of SOFT-TopK.

| $\tau = $ | 0.001 | 0.005 | 0.01 | 0.05 | 0.1 |
|---|---|---|---|---|---|
| objective score | 35956.6 | 42013.3 | **42520.8** | 41004.3 | 40721.2 |

Under the configuration used in our paper, both SOFT-TopK and GS-TopK have relatively good results. By grid search, we also discover better hyperparameters for both GS-TopK and SOFT-TopK, but it is worth noting that SOFT-TopK is still inferior to GS-TopK with the best-discovered hyperparameters. Our grid search result shows that our GS-TopK is not very sensitive to $\sigma$ if we have $\tau = 0.05$ or $0.1$, and the result of $\tau = 0.01$ is inferior because the Sinkhorn algorithm may not converge. The results of $\#G = 1000$ are all better than $\#G = 800$, suggesting that a larger $\#G$ is appealing if we have enough GPU memory. It is also discovered that SOFT-TopK seems to be able to accept a smaller value of $\tau$ compared to GS-TopK, probably because adding the Gumbel noise will increase the divergence of elements thus performs in a sense similar to decreasing $\tau$ when considering the convergence of Sinkhorn.

