# OpenReview forum: "On Learning to Solve Cardinality Constrained Combinatorial Optimization in One-Shot: A Re-parameterization Approach via Gumbel-Sinkhorn-TopK"
_ICLR.cc/2022/Conference — ICLR 2022 Submitted_

### Official Review · Reviewer_mjiu · 2021-10-29

**Correctness:** 3
**Technical Novelty And Significance:** 3
**Empirical Novelty And Significance:** 2
**Recommendation:** 5
**Confidence:** 4

**Main Review:**

This submission studies the learning-based formulation of cardinality constrained combinatorial optimization using probabilistic methods, specifically using Gumble-Sinkhorn method. By introducing Gumble noise into the existing Soft-TopK (Xie et al., 2020) formulation, the authors claimed that the bound of the difference between approximate objective function value and the ground-truth value is tightened, compared to the results in Xie et al., 2020. The authors presented Gumbel-Sinkhorn-TopK (GS-TopK) and its application in TopK combinatorial optimization problems. Experimental results for max covering and k-means clustering were presented to demonstrate the claim.

Strengths:

1. The introduction of Gumble noise addresses potential issues using deep network based optimal transport for learning-based combinatorial optimization, in particular, the diverging gap of Soft-TopK prediction and the optimal combinatorial solution, when the sorted approximate probabilities (of assignment) at the boundary are the same.

2. Following the previous theoretical analyses in the literature, the authors provided the proofs of the derived bounds in the appendix.

3. Both simple illustration (Figure 1) and empirical evaluations (Tables 1 and 2) for max covering and k-means clustering have validated the claim that the proposed GK-TopK improves Soft-TopK and outperforms other greedy and combinatorial optimization solutions, with respect to the achieved approximate objective function values.

Weaknesses:

1. It is not clear how the proposed formulation may change the derived combinatorial solutions besides the tightened gap for objective function values. This is related to the computation of objective function values in Algorithms 2 and 3 (in appendix), which basically took the best derived objective function values among the sampled transport maps for the GK-TopK formulation. Are the reported objective function values in Tables 1 and 2 based on these values? To access how this changes the solution quality, should the authors also provide the actual objective values with the derived combinatorial solutions? For now, it is difficult to tell whether the proposed method will have meaningful improvement in derived solution in practice.

2. It is not clear to me how significant is this proposed improvement over Soft-TopK. Based on the presentation, in particular, Theorem 1 and Remarks on page 6, by introducing Gumble noise, the derived bounds now have two terms. It may address the diverging gap issue when the sorted probabilities at the boundary are the same ($x_K = x_{K+1}$). But why when GS-TopK will always have tighter bounds than Soft-TopK even when $x_K \neq x_{K+1}$? Why adding Gumble noise to the original optimal transport formulation of Soft-TopK can always improve the bounds? In Figure 1, I assume that the actual optimal objective value is 2.4 while Soft-TopK has the gap at 2.0. What is the actual objective value for the derived solution by Soft-TopK? What about GS-TopK? Also, it seems the choice of $\tau$ does not affect the gap by Soft-TopK in this simple illustrative example but it does change GS-TopK gaps. But $\tau$ appears in the derived bounds for both formulations. Why?

3. For real applications, will different $\tau$ and $\sigma$ change the performances significantly?

4. Mathematical notations and the proofs have numerous issues. Here are a few examples: 1) In equation (6), $x_k$ and $x_{k+1}$ have not been defined yet. They were explained until the beginning of page 5. 2) Below equation (13) and the text after that, should "the optimal solution to Eq. (12)" $\tilde{T}$ be $\tilde{T}*$ instead? 3) In equations (17) and (18), is $\mathbf{x}_i$ a vector? Please make sure about the meaning of bold and regular fonts in these equations. Also, regular font $x$'s have been used for sorted probabilities. Different notations may need to be used to avoid confusion here. 4) In Appendix A.1 above equation (30), "By Lemma 1..." but this part is to prove Lemma 1. Do the authors mean by "Lemma 3" instead? Even that is the case, there appear to lack some steps to connect Lemma 3 with equation (30). 5) On page 19 above "Condition 1", "... except for the following condition: $x_i = x_k, x_i= x_k$." There must be typos for these two same equations. 6) Figure 4 and the proof for Theorem 2, do we need to worry about the cases with $x_i \leq x_j$ for Conditions 3 and 4?

5. There are missing information. For example, it is not clear what sample size #G was used in the experiments. It is not clear how exactly shrinking procedure was scheduled for homotopy GS-TopK as there is no detail provided in Algorithm 2. It would be also nice to discuss more why in Figure 3, the approximate formulations always have lower bounds. The lemmas, theorems or their proofs do not seems to indicate that will always happen?

6. Presentation can be improved significantly. For example, Figure 2 appears to be a general illustration instead of being the "overview of our pipeline" for max covering as indicated on page 6. There are also numerous typos, for example, "... important **hcardinality** constrained ..." in the last line of the paper on page 9.







**Summary Of The Paper:**

This submission studies the learning-based formulation of cardinality constrained combinatorial optimization using probabilistic methods, specifically using Gumble-Sinkhorn method. By introducing Gumble noise into the existing Soft-TopK (Xie et al., 2020) formulation, the authors claimed that the bound of the difference between approximate objective function value and the ground-truth value is tightened, compared to the results in Xie et al., 2020. The authors presented Gumbel-Sinkhorn-TopK (GS-TopK) and its application in TopK combinatorial optimization problems. Experimental results for max covering and k-means clustering were presented to demonstrate the claim.

**Summary Of The Review:**

The authors presented Gumbel-Sinkhorn-TopK (GS-TopK) to improve Soft-TopK. But there are concerns on the meaningful improvement for the actual derived approximate optimal solutions. The presentation of the submission can be improved significantly.

---

> ### Author Response · Authors · 2021-11-20
> **Response to Reviewer mjiu [2/2]**
>
> **Q5: For real applications, will different $\tau$ and $\sigma$ change the performances significantly?**
>
> - To address your concern, we update the paper with an ablation study in Table 3-5 in the supplementary material (Page 21-22). Our ablation study shows that our GS-TopK is not very sensitive to $\sigma$ if we have $\tau=0.05$ or $0.1$, and the result of $\tau=0.01$ is inferior because the Sinkhorn algorithm may not converge within 100 iterations. Generally, we have the following conclusions when setting the hyperparameters:
>
>   - **Setting $\tau$:** $\tau$ controls the gap of the continuous Sinkhorn solution to the discrete solution, and a smaller $\tau$ will lead to a tightened gap. This property is validated by our theoretical findings in Lemma 2. Unfortunately, $\tau$ cannot be arbitrarily small, because a smaller $\tau$ requires more Sinkhorn iterations to converge. Therefore, given a fixed number of Sinkhorn iterations (100) to ensure the efficiency of our algorithm, we need trial-and-error to discover the suitable $\tau$ for both SOFT-TopK and GS-TopK. The grid search results below show that our selection of $\tau$ fairly balances the performances of both SOFT-TopK and GS-TopK.
>   - **Setting $\sigma$:** As derived in Theorem 1, we need to balance between the two gaps Lemma 1 and Lemma 2) by a suitable $\sigma$. Since $|x_{k} - x_{k+1}|$ is unknown, we cannot directly find the theoretically optimal $\sigma$ given $\tau$. In the experiments, we firstly determine a $\tau$, and then find a suitable $\sigma$ by greedy search on a small subset ($\sim$5) of problem instances.
>
>   In ablation study, we choose the k=50, m=500, n=1000 max covering problem, and we have the following results (higher is better, we mark the configuration used in Table 1 by italic number):
>
>   **GS-TopK**
>
>   |               | $\\#G=1000$ |             |             | $\\#G=800$  |             |            |
>   | ------------- | ----------- | ----------- | ----------- | ----------- | ----------- | ---------- |
>   |               | $\tau$=0.01 | $\tau$=0.05 | $\tau$=0.1  | $\tau$=0.01 | $\tau$=0.05 | $\tau$=0.1 |
>   | $\sigma$=0.1  | 42513.4     | 44759.2     | **45039.5** | 42511.6     | 44754.6     | 45037.6    |
>   | $\sigma$=0.15 | 41456.5     | *44710.3*   | 44837.2     | 41421.4     | 44705.8     | 44841.5    |
>   | $\sigma$=0.2  | 41264.3     | 44638.1     | 44748.2     | 41235.9     | 44651.5     | 44748.6    |
>
>   **SOFT-TopK**
>
>   | $\tau=$   | 0.001   | 0.005   | 0.01        | 0.05      | 0.1      |
>   | --------- | ------- | ------- | ----------- | --------- | -------- |
>   | SOFT-TopK | 35956.6 | 42013.3 | **42520.8** | *41959.9* | 40721.17 |
>
>
>
> **Q6: Why in Figure 3, the approximate formulations always have lower bounds?**
>
> - To better clarify this, some remarks are added: the self-supervised learning pipeline in our GS-TopK, SOFT-TopK (Xie et al., 2021) and EGN (Karalias & Loukas, 2020) can be viewed as optimizing the objective score of the combinatorial optimization problem, yet the discrete constraints are relaxed to enable gradient-based optimization. Since the discrete clustering problem in Figure 3 is a minimization problem, relaxing the discrete constraints will allow the approximate formulations to find lower bounds.
>
>
>
> **Q7: About typos and minor misunderstandings.**
>
> - We truly appreciate the generous and detailed suggestions from the reviewer and we have updated the paper accordingly. Besides, there seems to be some minor misunderstandings towards our paper, which we also clarify and discuss here.
>
>   - On Page 4, $x_k, x_{k+1}$ are defined above Eq (7), so it seems to be a minor misunderstanding;
>   - On Page 5 below Eq (14), we fix the typo of $\widetilde{\mathbf{T}}^*$;
>   - On Page 7 below Eq (18), we update the notation of $\mathbb{I}(\mathbf{x})$;
>   - On Page 14, Section A.1, we fix the cross-reference to be Lemma 3, and we also update the intermediate steps in the proof of Lemma 1;
>   - On Page 19 above Condition 1, we fix the typos of $x_i = x_{k}$, $x_j = x_{k+1}$;
>   - On Page 19 in the proof of Theorem 2, we have covered the condition $x_i < x_j$ in Condition 2, thus it needs not be discussed in Condition 3 and 4. The condition $x_i = x_j$ can be covered by Condition 3 and 4, because we assume the values of $x_{k}, x_{k+1}$ are unique (if not, as discussed in the footnote on Page 18, the conclusion in Theorem 2 only changes by a constant);
>   - On Page 20, in Appendix C we update the configuration of $\\#G$ used in experiments.
>   - The configuration of Homotopy GS-TopK was originally provided on the top of Page 8, and now it is moved to Page 20, Appendix C due to limited pages.
>   - On Page 7 we update the description of Figure 2 as a "general illustration" of our pipeline;
>   - On Page 9 we fix the typo of "cardinality".

---

> ### Author Response · Authors · 2021-11-20
> **Response to Reviewer mjiu [1/2]**
>
> We would like to express our sincere gratitude for your detailed comments, and here we would like to clarify certain misunderstandings and list the detailed revisions made to our paper according to your precious suggestions. We are always available for your further queries.
>
> **Q1: Are the reported objective function values in Tables 1 and 2 based on the sampled transport maps? To access how this changes the solution quality, should the authors also provide the actual objective values with the derived combinatorial solutions?**
>
> - It seems to be a misunderstanding. The objective function values in Table 1 and 2 are based on the derived discrete combinatorial solutions, i.e. the discrete solutions constructed by firstly sorting the transportation maps, and then selecting the K-largest elements. Therefore we believe our result reported in experiments corresponds to meaningful improvement in practice. If there are any further queries, feel free to let us know so that we can better address your concerns.
>
>
>
> **Q2: What if $x_{k} \neq x_{k+1}$ in the derived bounds?**
>
> - The diverging gap is also critical if $x_k \neq x_{k+1}$ and $|x_k - x_{k+1}|$ is small. Theoretically, by referring to the mathematical form derived by Xie et al., 2020:
>
>   $$||\mathbf{T}^{*} - {\mathbf{T}}^{\tau *}||\_F \leq \frac{2 m\tau \log2}{|x_{k} - x_{{k+1}} |}$$
>
>   Even if $x_k \neq x_{k+1}$, given a small $|x_k - x_{k+1}|$ in the denominator, the gap will also become very large, such that SOFT-TopK (Xie et al., 2020) will fail to get a near-discrete solution which is crucial for an accurate estimation of the combinatorial objective score. In contrast, our bound has $\sigma^2$ in the denominator, thus the diverging issue will not be encountered if $\sigma$ is large enough. To better illustrate this issue, we update the toy example in Figure 1 by selecting the Top3 items from the following sequence: [1.0, 0.8, 0.601, 0.6, 0.4, 0.2], where there is a deterministic Top3 result [1.0, 0.8, 0.601]. However, as shown in the updated Figure 1, the improvement of the Gumbel trick is still significant under this scenario. (Since we cannot insert figures in openreview, please refer to Figure 1 in our updated paper.)
>
>
>
> **Q3: Why GS-TopK has tighter bounds than Soft-TopK?**
>
> - The bound of GS-TopK will be tighter than SOFT-TopK if we set a proper $\sigma$, and the effect of tighter bound will be more significant under the situation that $|x_k-x_{k+1}|$ is small, which is unavoidable for neural networks. SOFT-TopK can be viewed as the special case of GS-TopK by setting $\sigma=0$. For a sufficiently large $\sigma$, the bound derived in Lemma 2 is dominated by $\frac{\tau \log \sigma}{\sigma}$: if the other parameters are fixed, the bound will be tightened given a larger $\sigma$, thus GS-TopK will have tighter bounds than SOFT-TopK in the sense of Lemma 2 (i.e. the gap to a discrete solution). However, it is also worth noting that the bound in Lemma 1 is enlarged by a larger $\sigma$. Therefore, in practice, we have to balance between these two terms by selecting a suitable $\sigma$ in experiments.
>
>
>
> **Q4: It seems the choice of $\tau$ does not affect the gap by Soft-TopK in this simple illustrative example but it does change GS-TopK gaps. But $\tau$ appears in the derived bounds for both formulations. Why?**
>
> - Under the extreme case $|x_k - x_{k+1}| = 0$, the gap of SOFT-TopK becomes infinite thus it is no longer relevant to $\tau$. With our updated toy example in Figure 1 where $|x_k - x_{k+1}| = 0.01$, now a smaller $\tau$ will tighten the bound of SOFT-TopK, but it is still inferior compared to our GS-TopK with Gubmel noises.

---

### Official Review · Reviewer_fJUe · 2021-11-01

**Correctness:** 3
**Technical Novelty And Significance:** 3
**Empirical Novelty And Significance:** 3
**Recommendation:** 6
**Confidence:** 3

**Main Review:**

Overall, I enjoyed the paper. It introduces a simple trick, provides nice theory to justify the technique, and contains several experiments. I have a few comments.


- It would be nice to include a bit more math up front in Sections 1/2/3 about the combinatorial optimization problem being solved. Specifically in Section 3.1, $\mathbf{s}$ is not introduced, making it confusing to jump into the problem.
- I am confused that the proposed methods beat Gurobi/SCIP. I assume this is because of the runtime limit on the solver (100 sec). If so, I find 100 sec arbitrary. It would be more useful to let the solvers run to completion and report those results in addition to the current—I would describe that setting as “accurate” Gurobi/SCIP rather than the current.
- As a result, it is also difficult to say that the current approach “outperforms state-of-the-art Gurobi” without specifying that this is under runtime constraints. Outperforming the solvers would make sense if the current methods can find *provably optimal* solutions faster than the solvers, or if the solvers cannot obtain solutions within their understanding of reasonable time (e.g., ~hours)
- It is hard to determine the effectiveness of the approach by comparing raw objective function value. Given a full-runtime globally optimal solution, we can then evaluate in terms of the relative suboptimality gap of each method. This would give a clearer picture of how much better the current method is over baselines.
- I am surprised that the discontinuity in the optimality gap can incur such a high sub-optimality, since it seems like a small problem. I appreciate the numerical toy example, but it is still not fully clear. Is the reason because in training $x_k$ and $x_{k+1}$ can repeatedly cycle in training, e.g., one increases over the other then the other increases in each iteration? A bit more intuition to the reasoning would be appreciated.


**Summary Of The Paper:**

This paper advances one-shot solution generation methods for cardinality constrained optimization problems. The recently developed SOFT-TopK algorithm suffers from an optimality bound discontinuity. This paper introduces a Gumbel randomization trick to the algorithm, prove that this removes the discontinuity in the bound, and implement their algorithm in two combinatorial problems, where they demonstrating achieving better solutions in less time than baselines.


**Summary Of The Review:**

I emphasize that I like the paper and my main concerns are on fair comparisons. I would be happy to increase my score if the authors include an additional Gurobi/SCIP baseline where they let the solvers run for sufficiently long to solve the problems to global optimality, and then compare against the relative optimality gap from the global solution.

---

> ### Author Response · Authors · 2021-11-20
> **Response to Reviewer fJUe**
>
> Many thanks for identifying the novelty of our paper and your precious comments are well appreciated. Here we set out below our responses to your valuable questions.
>
> **Q1: About comparing with the optimal solutions.**
>
> - According to your kind suggestions, we remove the time limit and let Gurobi solve the globally optimal solutions. We update Table 2 with the optimal objective scores and the gap to optimal objectives for the discrete clustering problem. Our method takes less than 1/3 time cost compared to the optimal Gurobi solvers, and the approximation ratios are around 2%. The speedup is more significant for the larger-sized problem. We also try to acquire the optimal result by Gurobi for the max covering problem, however, the solver does not reach the optimal solution within 24 hours for a single problem instance thus we report an out-of-time (OOT) in Table 1.
>
>
>
> **Q2: A bit more intuition about why the discontinuity in the optimality gap can incur such a high sub-optimality.**
>
> - According to our analysis, the improvement achieved by our GS-TopK is mainly because we can reach a more accurate estimation of the objective score during training. To better clarify this, some remarks are added: the self-supervised learning pipeline in our GS-TopK, SOFT-TopK (Xie et al., 2021) and EGN (Karalias & Loukas, 2020) can be viewed as optimizing the objective score of the combinatorial optimization problem, yet the discrete constraints are relaxed to enable gradient-based optimization. Since the decision variables are continuous, the estimation of the objective score may be inaccurate, due to the gap between the continuous solution and its MAP-inference discrete solution. Our theoretical analysis characterizes such a gap, and we prove that our gap improves SOFT-TopK by incorporating the Gumbel trick. To conclude, the connection between our theoretical findings and the experiment results is that our GS-TopK can estimate the objective score more accurately because our derived gap to the MAP-inference is tightened compared to SOFT-TopK. Such a connection is also validated by Figure 3 in our paper.
>
>
>
> **Q3: About typos, revisions, and minor misunderstandings.**
>
> - We truly appreciate your suggestions. Our modifications to the paper are summarized here concerning each of your suggestions. And there seem to be some minor misunderstandings, which we also clarify here.
>
>   - We update Eq (1) on Page 1 to include the general mathematical formulation of cardinality constrained combinatorial problems which can be tackled by our proposed method.
>   - There seems to be a misunderstanding in Section 3.1, as we have introduced $\mathbf{s}$ on Page 3 above Eq (2): "for selecting the k largest items from the probability vector s".
>   - On Page 9, we update the names of SCIP and Gurobi in Table 1 and 2 into SCIP/Gurobi (faster/slower) to clarify that these solvers are with time constraints. We also include the Gurobi (optimal) entry as per your precious suggestion.
>   - On Page 1 and 2, we emphasize in the Abstract and the Introduction Section that the SCIP and Gurobi solvers are with time constraints, and our approach achieves better solutions with less consumed time.

---

### Official Review · Reviewer_J7pw · 2021-11-02

**Correctness:** 3
**Technical Novelty And Significance:** 3
**Empirical Novelty And Significance:** 2
**Recommendation:** 5
**Confidence:** 4

**Main Review:**

The paper proposes a new application for differentiable top-k as well as a variation of a differentiable top-k operator that improves theoretical guarantees, which they also prove.

There are also other soft top-k operators.
Very influential for the work by Xie et al. is the optimal transport based differentiable sorting and ranking (https://arxiv.org/abs/1905.11885).
There are also differentiable sorting networks, which provide differentiable sorting and ranking (https://arxiv.org/abs/2105.04019).
Both of these works include differentiable top-k operators and include experiments demonstrating their utility.
I suggest adding and discussing these references and elaborating the choice of using the formulation of Xie et al.
(At the moment, it seems like the paper implies that the choice was because this would be "the" or the only soft top-k operator, which is not the case.)

In the toy example in Figure 1, there is not a unique solution of the top-k operation.
Therefore: why is it beneficial that one specific solution is selected over a 50% / 50% mixture of both 0.6 entries?
The algorithm 2 would result in a 50% / 50% mixture for the two 0.6 entries.
Differentiable sorting networks for top-k achieve a 50% / 50% mixture without sampling.
This should be discussed.

In algorithm 3, there is a $T$ in line 7. The $ \mathbf{\tilde T}_i$s are not used, so maybe there is a typo?

The citation for "The Concrete Distribution: A Continuous Relaxation of Discrete Random Variables" by Maddison et al. is missing, which was published simultaneously to Jang et al. Please add it respectively.

Is the difference between the "SOFT-TopK w/ sampling" method and the "GS-TopK" method that sampling for "SOFT-TopK w/ sampling" is done only during inference, while for "GS-TopK" sampling is done during both inference and training?

What are the differences to the Gumbel-Sinkhorn method by Mena et al.? From "However, these existing methods do not study the application in combinatorial optimization learning, and the gap between the MAP inference and the result after Gumbel re-parameterization is not well characterized in (Mena et al., 2018).", I understand that they are fairly similar, and the main difference is application and theoretical guarantees.

I did not check the validity of the theoretical results.

I appreciate the application and experiments. However, I wonder how the hyper-parameters were selected.
From the current draft, it seems like an "empirical selection" was done for your proposed method, but there is not a clear procedure to have a meaningful comparison between all methods.
Optimizing only your hyperparameters would lead to a significant bias in the results.
There is no grid specified over which you perform grid search to give each method a fair chance.
At a time cost of seconds for each method, it should be very feasible to do even a quite extensive grid search.


### TYPOs:
Overall, the grammar should be reworked, e.g., using a tool like LanguageTool or Grammarly, would point out various typos that I did not cover below.

p.2: and pipeline is differentiable -> and the / our pipeline is differentiable

p.2: taxonomy by (Peng et al., 2021) -> taxonomy by Peng et al. (2021)    (This TYPO is also at other locations, e.g., Inspired by
(Jang et al., 2017),)

p.3: parametreization.

gradient ascend -> gradient ascent

p.7 problems pf cardinality -> problems of cardinality

p.9 Directly applying Gumbel sampling to SOFT-TopK seems either an effective improvement, and satisfying
results are achieved when exploiting the Gumbel re-parameterization trick and enabling gradient-
based optimization over the combinatorial objective score.  (This sentence is grammatically incorrect, which makes it not understandable.)

at various occasions, you write SOTF-TopK instead of SOFT-TopK

**Summary Of The Paper:**

The paper proposes to add Gumbel noise to the scores to which differentiable top-k via regularized optimal transport is applied.
This improves theoretical guarantees for the case where the $k$th and the $k+1$th elements are equal.

**Summary Of The Review:**

Overall, I see that the paper has theoretical strengths and presents an interesting application for the method.
However, the methodological contribution is small.
The experimental evaluation is or seems to be flawed, as it is questionable whether the comparison is fair.
Grammatically, the paper should be reworked. (which does not impact my score, as I expect this to be fixed in the final paper.)

For these reasons, I suggest rejecting this work in its current form.

Improving and extending the comparison to related work (e.g., Mena et al.) and making the comparison explicit would improve the paper. It is important to explicitly state the differences if there are any.
The empirical evaluation should be more rigorous.

---

I appreciate the response and revision; however, I still have concerns as per my last response.

---

> ### Comment · Reviewer_J7pw · 2021-11-18
> **Comment**
>
> As you have not responded to my review, I maintain my score for now.
>
> **Update:** Thanks for your response. Due to other obligations, it will take at least a few days until I will be able to read it carefully (just so you are informed).

---

> ### Author Response · Authors · 2021-11-19
> **Response to Reviewer J7pw [3/3]**
>
> **Q5: About the differences between this paper and Mena et al. 2018: I understand the main difference is application and theoretical guarantees.**
>
> * Many thanks for comparing our paper with Mena et al., 2018, a great milestone in our study field. Although we share certain technical components with this paper, we would like to illustrate that we have made some unique contributions in our paper:
>
> 	**Application**. Firstly, as identified by other reviewers, the experiment results achieved in our application (combinatorial optimization) are non-trivial. We outperform state-of-the-art commercial solver Gurobi given a smaller time budget, and we also outperform the state-of-the-art one-shot learning algorithm EGN (Karalias & Loukas, 2020). Besides, the application of combinatorial optimization learning is a trending research topic in ICLR/ICML/NeurIPS, however, most existing papers are focused on building reinforcement learning models. Novel ML tools such as optimal transport and the Gumbel trick are underexploited for combinatorial optimization, and we feel passionate that this paper will inspire more researchers in developing OT and Gumbel-based combinatorial optimization learning methods in the future.
>
> 	**Theoretical guarantees.** We would also like to emphasize that our theoretical derivations are non-trivial, by theoretically decomposing the gap of GS-TopK into two terms (bias by the Gumbel noise plus the gap to a discrete solution). The hyperparameter selection procedure is also guided by our theoretical derivations (see details in Q4). And it is worth noting that there are just few papers with theoretical results within the scope of machine learning for combinatorial optimization, which makes our paper very unique in this field. Finally, our theoretical derivation may serve as a complement for Mena et al., 2018.
>
> **Q6: About Typos and Suggested References.**
>
> * Your valuable comments are well received and we have updated the paper accordingly:
>   * See Page 20 for the updated Algorithm 3 with fixed typos;
>   * See Page 3 for your suggested reference.

---

> ### Author Response · Authors · 2021-11-19
> **Response to Reviewer J7pw [2/3]**
>
> **Q4: I wonder how the hyper-parameters were selected. Optimizing only your hyperparameters would lead to a significant bias in the results. Are you optimizing only your hyperparameters by a quite extensive grid search?**
>
> * Optimizing only our hyperparameters by extensive grid search is the approach that we definitely will not consider as it is against our academic moral rules. In our paper, we optimize the hyperparameters by greedy search on a small subset of problem instances (~5) and set the best configuration of hyperparameters for both GS-TopK, SOFT-TopK. Our search of hyperparameters is also based on our theoretical findings. The following three hyperparameters are crucial to the performance of GS-TopK, and $\tau$ is also crucial for SOFT-TopK, and we set the same $\tau$ for both GS-TopK and SOFT-TopK in the sense of fair comparison. We would like to take this opportunity to add some remarks here:
>
> 	* **$\\#G$ (number of Gumbel samples)**: $\\#G$ affects how many samples are taken during training and inference for GS-TopK. A larger $\\#G$ (i.e. more samples) will be more appealing, because GS-TopK will have a more accurate estimation of the objective score, and it will have a higher probability of discovering better solutions. However, $\\#G$ cannot be arbitrarily large because we are running GS-TopK in parallel on GPU, and the GPU has limited memory. Therefore, in experiments, we set a large enough $\\#G$ (e.g. $\\#G=1000$) and ensure that it can fit into the GPU memory of our workstation (2080Ti, 11G).
> 	* **$\tau$ (entropic regularization factor of Sinkhorn)**: Theoretically, $\tau$ controls the gap of the continuous Sinkhorn solution to the discrete solution, and a smaller $\tau$ will lead to a tightened gap. This property is validated by our theoretical findings in Lemma 2. Unfortunately, $\tau$ cannot be arbitrarily small, because a smaller $\tau$ requires more Sinkhorn iterations to converge. Therefore, given a fixed number of Sinkhorn iterations (100), we need trial-and-error to discover the suitable $\tau$ for both SOFT-TopK and GS-TopK. The grid search results below show that our selection of $\tau$ fairly balances the performances of both SOFT-TopK and GS-TopK.
> 	* **$\sigma$ (Gumbel noise factor)**: As derived in Theorem 1, we need to balance between the two gaps (Lemma 1 and Lemma 2) by a suitable $\sigma$. Since $|x_{k} - x_{k+1}|$ is unknown, we cannot directly find the theoretically optimal $\sigma$ given $\tau$. In the experiments, we firstly determine a $\tau$, and then find a suitable $\sigma$ by greedy search on a small subset (~5) of problem instances.
>
> 	__Ablation study: Grid search of hyperparameters.__
>
> 	With the great inspiration gained from your precious comments, we conduct an ablation study about the sensitivity of hyperparameters by performing an extensive grid search near the configuration used in our experiments. We choose the k=50, m=500, n=1000 max covering problem, and we have the following results (higher is better), and we mark the result in our main paper by italic number:
>
> 	**GS-TopK**
>
> 	|   | $\\#G=1000$ | | | $\\#G=800$ | | |
> 	| ----- | ------ | ---- | ---- | --- | --- | --- |
> 	|   | $\tau$=0.01 | $\tau$=0.05 | $\tau$=0.1  | $\tau$=0.01 | $\tau$=0.05 (in paper) | $\tau$=0.1  |
> 	| $\sigma$=0.1  | 42513.4 | 44759.2  | **45039.5** | 42511.6 | 44754.6  | 45037.6 |
> 	|  $\sigma$=0.15 | 41456.5 | *44710.3* | 44837.2 | 41421.4 | 44705.8  | 44841.5     |
> 	|  $\sigma$=0.2 | 41264.3  | 44638.1 | 44748.2 | 41235.9 | 44651.5  | 44748.6     |
>
> 	**SOFT-TopK**
>
> 	| $\tau=$   | 0.001   | 0.005   | 0.01  | 0.05 | 0.1 |
> 	| -- | --- | ---- | ---- | --- | --- |
> 	| SOFT-TopK | 35956.6 | 42013.3 | **42520.8** | *41959.9* | 40721.17 |
>
> 	**Our analysis:** under the configuration used in our paper, both SOFT-TopK and GS-TopK have relatively good results, thus we do not find a set of hyperparameters to intentionally poison the performance of SOFT-TopK. By grid search, we also discover better hyperparameters for both GS-TopK and SOFT-TopK, but it is worth noting that SOFT-TopK is still inferior to GS-TopK with the best discovered hyperparameters. Our grid search result shows that our GS-TopK is not very sensitive to $\sigma$ if we have $\tau=0.05$ or $0.1$, and the result of $\tau=0.01$ is inferior because the Sinkhorn algorithm may not converge. The results of $\\#G=1000$ are all better than $\\#G=800$, suggesting that a larger $\\#G$ is appealing if we have enough GPU memory. It is also discovered that SOFT-TopK seem to be able to accept a smaller value of $\tau$ compared to GS-TopK, probably because adding the Gumbel noise will increase the divergence of elements thus performs in a sense similar of decreasing $\tau$ when considering the convergence of Sinkhorn. Since we have discovered a set of better hyperparameters, we are going to validate its effectiveness on other settings and update the result in the camera-ready version accordingly.

---

> > ### Comment · Reviewer_J7pw · 2021-11-27
> > **Response**
> >
> > Thank you again for your response.
> >
> > I would like to clarify my concerns regarding the hyper-parameters: You compare to many methods, specifically: kmeans, kmeans++, EGN, SCIP, Gurobi. Do these methods have hyperparameters? If yes, which grid did you use? (I do not mean display results for all hyperparameter configs but just specifying the exact grid you used, or a list of the specific setting that you did consider.)
> > If they do not have user specifiable hyperparameters. (I do not have experience, e.g., in Gurobi, but from experience in similar software, I would expect that it uses some kind of heuristic to specify internal hyperparameters potentially inaccessible by the user.) If running multiple settings with your method, while only one specific setting is and can be used by the baseline, this induces an implicit bias.
> >
> > Another concern about runtime comparison: Are the baseline methods executed on CPU, and your methods on GPU?
> >
> > Could you please clarify this? Thanks.

---

> > > ### Author Response · Authors · 2021-11-28
> > > **Clarification about hyperparameters**
> > >
> > > We truly appreciate the reviewer for your updates, and here are our clarifications about hyperparameters.
> > >
> > > 1. About whether the other methods have hyperparameters.
> > >     - **kmeans** and **kmeans++** are classic clustering methods and do not have hyperparameters.
> > >     - **greedy** is the greedy algorithm for max covering and does not have hyperparameters.
> > >     - **EGN** is a self-supervised learning pipeline proposed by (Karalias and Loukas, 2020), and it has two hyperparameters the learning rate $lr$ and the balance between the constraint and the objective score $\beta$. In our experiment, we perform a greedy search of hyperparameters which is in line with ours, with $lr=(0.1, 0.01, 0.001, 0.0001)$ and $\beta=(0.1, 0.5, 1, 2, 10)$, and we set $lr=0.001$, $\beta=10$ in our experiments. The inferior result for EGN, according to our analysis, is because that it does not encode the constraint in the neural network output.
> > >     - **Gurobi** and **SCIP** have many hyperparameters, and we adopt the default configurations as the baseline. However, I would like to point out that the default hyperparameters of Gurobi and SCIP are tested and configured by a team of full-time employees (for Gurobi), or an active community of operations research experts (for SCIP). It is non-trivial to surpass the default configuration, and reviewer BoAG describes our experiment as "strong experiment results over baselines". Therefore, we believe that Gurobi and SCIP can serve as a strong enough baseline under their default confugirations, and existing learning-based combinatorial optimization papers (Karalias and Loukas, 2020, Dai et al., 2017) also compare with the default configuration of combinatorial optimization solvers.
> > >
> > > 2. About running on CPU or GPU:
> > >     - **kmeans, kmeans++** run on GPU (for comparing the pair-wise distance and doing the argmin step);
> > >     - **greedy** runs on CPU because we do not find any critical steps that can be accelerated by GPU;
> > >     - **EGN** runs on GPU which is in line with the original paper (Karalias and Loukas, 2020);
> > >     - **Gurobi** and **SCIP** run on CPU because these branch-and-bound methods contain too many discrete decision steps, therefore, cannot exploit the SIMD nature of GPU. However, Gurobi will take the advantage of the multi-core CPU during computation. In comparison, one of the advantages of learning combinatorial optimization methods, as also analyzed by other papers (Karalias and Loukas, 2020, Wang et al., 2021), is that learning methods can exploit the advantage of GPU, but existing CPU-based solvers can not.
> > >
> > > Since we cannot revise the paper right now, we will update the above information in the final version. If the reviewer has any further concerns, we will be ready to respond at our best effort.
> > >
> > > **References**
> > >
> > > (Karalias and Loukas, 2020) Nikolaos Karalias and Andreas Loukas. Erdos goes neural: an unsupervised learning framework for combinatorial optimization on graphs. In Neural Info. Process. Systems, 2020.
> > >
> > > (Dai et al., 2017) Hanjun Dai, Elias Khalil, Yuyu Zhang, Bistra Dilkina, and Le Song. Learning combinatorial optimization algorithms over graphs. In Neural Info. Process. Systems, pp. 6351–6361, 2017.
> > >
> > > (Wang et al., 2021) Runzhong Wang, Junchi Yan, and Xiaokang Yang. Neural graph matching network: Learning
> > > lawler’s quadratic assignment problem with extension to hypergraph and multiple-graph matching. Trans. Pattern Anal. Mach. Intell., 2021.

---

> ### Author Response · Authors · 2021-11-19
> **Response to Reviewer J7pw [1/3]**
>
> We truly appreciate your detailed feedbacks on our paper, and our sincere apologies for not being able to reply with our responses earlier, as we are working intensively towards the result of additional experiments as per your suggestions, in the wish to deliver with consistent and clean responses.  However, we notice that there might be certain misunderstandings towards our paper. We set out below our responses to each of the questions and we remain available for any of your further queries.
>
> **Q1: The suggestion about other soft top-k operators.**
>
> * Many thanks for pointing out these two important soft top-k papers based on soft sorting techniques, and we have incorporated these two papers into the section of related works.
>
>   After carefully going through and thoroughly analyzing these papers, we would like to supplement with our remark and rationale on why we believe Xie et al., 2020 would be a more suitable choice within the scope of our paper (with the Gumbel trick and applications to combinatorial optimization): assuming $m$ items, the transportation matrix size of Xie et al., 2020 is $m \times 2$, but the transportation matrix size of the soft sorting papers is (at least) $m \times (k+1)$. Therefore, given a limit of GPU memory, SOFT-TopK (Xie et al., 2020) will allow $(k+1)/2$ more Gumbel samples compared to soft sorting papers. And our analysis (see discussion about $\\#G$ in Q4) shows that a larger number of Gumbel samples will lead to better optimization result for our GS-TopK.
>
>   Considering the above, we are of the view that that the papers that you have kindly pointed out for us have parallel contributions with SOFT-TopK  (Xie et al., 2020). And as we focus on the application in combinatorial optimization learning in our paper, we have the confidence to believe that we have made our own and unique technical and theoretical contributions.
>
> **Q2: In the toy example in Figure 1, there is not a unique solution of the top-k operation. Why is it beneficial to get a discrete solution?**
>
> * Firstly, we would like to clarify that a discrete solution is always being favored within the scope of this paper, because we are handling combinatorial optimization learning where the decision variables should be near-discrete to acquire an accurate estimation of the object score, and our GS-TopK has improved discreteness compared to SOFT-TopK (especially when $x_k-x_{k+1}$ is small, proven by Lemma 2), thus our SOFT-TopK provides a more accurate estimation of objective score in experiments (see Figure 3).
>
>   Secondly, by referring to the mathematical form derived by Xie et al., 2020:
>
>   $$||\mathbf{T}^{*} - {\mathbf{T}}^{\tau *}||\_F \leq \frac{2 m\tau \log2}{|x_{k} - x_{{k+1}} |}$$
>
>   Even if $x_k \neq x_{k+1}$, given a small $|x_k - x_{k+1}|$ in the denominator, the gap will also become very large, such that SOFT-TopK (Xie et al., 2020) will fail to get a near-discrete solution which is crucial for an accurate estimation of the combinatorial objective score. To better illustrate this issue, we update the toy example in Figure 1 by selecting the Top3 items from the following sequence: [1.0, 0.8, 0.601, 0.6, 0.4, 0.2], where there is a deterministic Top3 result [1.0, 0.8, 0.601]. However, as shown in the updated Figure 1, the improvement of the Gumbel trick is still significant under this scenario. (Since we cannot insert figures in openreview, please refer to Figure 1 in our updated paper.)
>
>   Hope the above is clear and please feel free to reach out to us if you would like to further discuss.
>
> **Q3: Is the difference between the "SOFT-TopK w/ sampling" method and the "GS-TopK" method that sampling for "SOFT-TopK w/ sampling" is done only during inference, while for "GS-TopK" sampling is done during both inference and training?**
>
> * That is exactly the point we would like to convey in our paper. And what we want to validate by the "SOFT-TopK w/ sampling" entry is that the improvement brought by GS-TopK is not simply due to the sampling effect of Gumbel noise in testing, but mainly because we can estimate the combinatorial objective score more accurately with the Gumbel noise during training. Such a phenomenon also validates our theoretical findings.

---

> ### Comment · Reviewer_J7pw · 2021-11-29
> **Thanks for the response.**
>
> Thanks for the response. I would like to apologize for my delayed response.
>
> "Q1: The suggestion about other soft top-k operators."
>
> I agree with your response and would appreciate if this discussion is included in the final version / supplementary material.
>
> "Q2: In the toy example in Figure 1, there is not a unique solution of the top-k operation. Why is it beneficial to get a discrete solution?"
>
> I understand.
>
> I have the concern that by perturbing with a Gumbel distribution, it could also happen that $|x_k-x_{k+1}|$ becomes smaller. Is there an intuition why this is not a problem?
>
> You are writing in your paper that the upper bound of the gap diverges.
> Initially reading, I had gotten the impression that the result would diverge, but the actual result does not diverge, right?
> Maybe you could keep this potential confusion in mind to clarify it for the revision/final version.
>
> When I was asking "What are the differences to the Gumbel-Sinkhorn method by Mena et al.?", I wanted you to clarify the exact methodological differences of your method; I think that would be important for the distinction to related work. I apologize if that was ambiguous. As I said, I was aware of the theoretical and application related differences.
>
> "Is the difference between the "SOFT-TopK w/ sampling" method and the "GS-TopK" method that sampling for "SOFT-TopK w/ sampling" is done only during inference, while for "GS-TopK" sampling is done during both inference and training?": Could you please clarify this inside the table, or in the caption etc. as it leads to guessing what the abbreviations actually mean?
>
> In table 2, you report "Gurobi 9.0 (optimal)" which seems to be the best method but is not marked in bold font. Why?
> Also, I disagree that the time difference is really that notable. Comparing 8 CPU cores (how many did you use for Gurobi?) to a full 2080 Ti is not really appropriate, and I would recommend computing both for a fair comparison on a single CPU core.
> Heuristically, I would suggest 100 cores being equivalent to a 2080ti. CPUs with more than 100 threads do exist, and I would have liked to see a comparison either on such a level or simply each method on a single CPU core.
> When comparing total system power consumption, I would also assume the Gurobi to have a better trade-off.
>
> TYPO: gradient descend -> gradient descent
>
> **Update**: I just realized that OOT might be due to out-of-memory / slow swap / recomputing discarded results, etc. I find 16 GB on a machine really small (I am sorry if this is all you can get a hold of). I would recommend trying some free trial of a cloud provider (I remember once having 300 USD free on Google cloud). This should be enough to get a ~96 thread + ~500GB/1TB ram machine for 100 hrs or so.

---

> > ### Author Response · Authors · 2021-11-30
> > **Some Clarifications of Minor Minsunderstandings**
> >
> > We truly appreciate the reviewer for the update, and here we would like to clarify some minor misunderstandings and respond to your further concerns.
> >
> > 1. By perturbing with a Gumbel distribution, it could also happen that $|x_k-x_{k+1}|$ becomes smaller. Is there an intuition why this is not a problem?
> >
> > - You may refer to our theoretical result for the answer to this problem: under the sense of expected gap over Gumbel noises, the gap in Lemma 2 will become smaller given a larger $\sigma$. Intuitively, we can say that it is more likely that $|x_k-x_{k+1}|$ will become larger because the Gumbel noise can be regarded as a distribution with sparse spikes, and it is very likely that $|x_k-x_{k+1}|$ will become larger if $x_k, x_{k+1}$ is affected by the spikes. (NOTE: This is a very informal and intuitive explanation, and we recommend referring to the theoretical result.)
> >
> > 2. What are the methodological differences between Mena et al., 2018?
> >
> > - There is a difference in the methodology: Mena et al. are focused on the permutations, i.e. their outputs are doubly-stochastic matrices, and they assume that the marginal distributions in optimal transport are uniform and do not explicitly consider non-uniform marginal distributions in their implementation. And our method is focused on top-k where the marginal distributions are non-uniform, and we implement the Gumbel-Sinkhorn algorithm with explicit treatment on non-uniform marginal distributions. We also believe that our non-trivial contributions in theory and applications are strong enough to distinguish our paper from Mena et al., 2018.
> >
> > 3. Gurobi 9.0 (optimal) seems to be the best method which is not marked in bold, why?
> >
> > - Gurobi 9.0 (optimal) serves for optimal values to compute the "optimal gap" for all methods, and it is not marked in bold because it consumes more time than other methods. Since solving combinatorial optimization means trading off between time and accuracy, it is a common practice for combinatorial optimization researchers to adopt Gurobi for optimal values, and do not consider it as a peer method for comparison. Please note that Reviewer fJUe agrees with us and has mentioned that "I think the empirical results of the paper (from Table 1 and Table 2) are very strong, especially the improvements over SOFT-TopK from 12% to 2% optimality gap." Please also refer to Table 1 in [Kool et al., 2019](http://arxiv.org/abs/1803.08475) which is a great milestone for learning combinatorial optimization, where the optimal results by Gurobi are either included for comparison. We will rearrange Tables 1 and 2 in the final version and add clarification to address your concerns.
> >
> > 4. About comparing with CPU/GPU for Gurobi and our method.
> >
> > - Firstly, we would like to clarify that the underlying algorithm of Gurobi, which is known as branch-and-bound, cannot be easily parallelized as you may expect because it involves sequential discrete decision steps when branching on the search tree.
> >
> >   Secondly, the current multi-threading strategy by Gurobi is setting out multiple independent solvers in parallel, and the improvement may not be as significant as you expected, because it still requires a single-thread solver to traverse the whole search tree before returning the optimal result. We refer to the [official support from Gurobi](https://support.gurobi.com/hc/en-us/articles/360013419951-Does-using-more-threads-make-Gurobi-faster-) to further clarify the concern: "For MIPs where the optimization requires a large number of nodes, more threads can improve the time to solve. For MIPs that are solved at or near the root node, more threads generally will not help much."
> >
> >   Thirdly, Gurobi consumes 4.5G RAM when solving the max covering problem, and we believe that the RAM size is not a bottleneck. We also have tested on a server with CPU E5-2678 CPU (48 cores, 2.50GHz) + 128GM RAM, while our workstation is with i7-9700k (8 cores, 3.60GHz) + 16GB RAM, and empirically Gurobi performs better on i7-9700k, which is reasonable because the single-thread performance seems to be more important than the number of cores for the branch-and-bound solvers.
> >
> >   Finally, we are sorry that it might be infeasible to test on cloud computing machines, because Gurobi is commercial software and we can only apply for a limited number of academic licenses for our own computers. It may even raise legal issues if we install an academic license on a cloud computing machine. We hope our discussion above can relieve your concerns, and we would like to highlight again that one of the advantages of machine learning for combinatorial problems is that machine learning methods can exploit the power of GPU but existing classic solvers cannot.
> >
> > We also appreciate the other detailed suggestions for our final version, and we will update the paper accordingly. We will be truly grateful if you can reconsider our paper, and we are ready to respond to any of your further concerns at our best effort.

---

> > > ### Comment · Reviewer_J7pw · 2021-11-30
> > > **Thank you.**
> > >
> > > Thank you for the clarifications.
> > >
> > > Could you include single-thread CPU results for the experiments in Table 1&2?
> > > While I agree that a comparison between the best possible hardware for each method makes sense, it does not paint the full picture.
> > > I think that for a fair comparison, the aspect of CPU runtime of each method is also important.
> > > Let's say someone wants to compute everything on CPU as they do not have an NVIDIA GPU, etc. Or someone wants to solve a large number of problems, then a CPU runtime would be the more important metric.
> > > I do expect the CPU runtime to be 10-100x slower than Gurobi, and I think that is ok, but I think it is a problem that this is not communicated in the paper.
> > > You present your method as being simply faster, which is problematic as it requires specialized graphics hardware and is most likely slower on general purpose hardware.
> > > I would like to see a discussion on this (including some CPU runtime results) in the final revision.
> > >
> > > "Finally, we are sorry that it might be infeasible to test on cloud computing machines," You may have misunderstood my comment, it was a suggestion in case you do not have larger hardware than the "workstation" you report in your paper.

---

> > > > ### Author Response · Authors · 2021-11-30
> > > > **Thanks for your kind suggestions**
> > > >
> > > > **Update:** We fix some typos below and add discussions for your easy reference.
> > > >
> > > > Many thanks for your timely update, and we agree that current machine learning methods, including our GS-TopK, rely on GPU for their high efficiency. Here we provide the timing of machine learning methods on CPU on the first test instance of max covering problem for a quick glimpse (the objective scores are taken from Table 1, higher is better):
> > > > * EGN (accurate, CPU) (Karalias & Loukas, 2020): obj=36927.3, time=348.154s
> > > > * SOFT-TopK (CPU) (Xie et al., 2020): obj=41959.9, time=291.748s
> > > > * SOFT-TopK (w/ sampling, CPU): obj=39684.8, time=1104.808s
> > > > * GS-TopK (CPU) (ours): obj=44710.3, time=603.211s
> > > >
> > > > And we list the time of Gurobi for your quick reference (faster/slower/optimal mean different time limits for Gurobi):
> > > > * Gurobi 9.0 (faster): obj=43261.6, time=30.078s
> > > > * Gurobi 9.0 (slower): obj=43937.2, time=100.171s
> > > > * Gurobi 9.0 (optimal): OOT
> > > >
> > > > It is worth noting that the objective scores solved by the above methods are different. For a fair comparison, we allow both GS-TopK and Gurobi to run 600s on CPU. For the first test instance, the objective score of GS-TopK is 44001, which is better than 43508 solved by Gurobi under 600s. Therefore, at least on the max covering problem, the performance of GS-TopK with CPU may not be as poor as you expected, and we may need some additional experiments (which may take several days) to conclude whether GS-TopK is inferior or better than Gurobi if both methods are run on CPU under the same time budget.
> > > >
> > > > Finally, we believe that the efficiency of most modern machine learning models should rely on the power of GPU. We will add discussions and update the detailed result in the final version. We hope these results can relieve your concerns about our paper.

---

### Official Review · Reviewer_BoAG · 2021-11-03

**Correctness:** 3
**Technical Novelty And Significance:** 2
**Empirical Novelty And Significance:** 3
**Recommendation:** 6
**Confidence:** 4

**Main Review:**

## Strengths:
- Strong experimental results with baselines that include other neural approaches as well as solvers and classical algorithms.
- The paper builds on recent work from the literature.
- The approach provides improved theoretical results for soft TopK through the use Gumbel noise.
- The motivation and the explanation of the theoretical results are clear and well done.

## Comments and weaknesses:
- Minor corrections regarding related work.
You claim "Besides, Paulus et al.
(2020) develop a general framework for generating combinatorial distributions via Gumbel distribution." The work in Paulus et al. can use different
kinds of noise in their optimization program, not just Gumbel.
Regarding the work by Karalias and Loukas, you claim "However, they do not consider the cardinality constraint (at least not
explicitly)." While the paper indeed does not explicitly address cardinality constraints in the examples/experiments, it is fairly straightforward to describe such constraints with their probabilistic penalty approach.

- Just below Equation 13, the optimal solution is denoted with a tilde. Is an asterisk missing there or am I misunderstanding something?
The notation is somewhat confusing at that point. The expectation on the lemma right below has T with both tilde and an asterisk but the
sentence claims that the optimal is just tilde(T) (well, boldface T to be precise).
More generally, the text contains various typos that need to be fixed, for example:
page 5: "our aim becomes to proof the upper bound"
page 7: "learning two representative problems pf cardinality".


- Regarding the motivation of the paper, the authors explain ties between $x_k$ and $x_{k+1}$ will break the original bound, which is a concern
because those are outputs of a neural network so it is a plausible scenario. Is this really an issue in practice? That is, does the Gumbel noise
end up breaking ties that appear in neural network outputs in your experiments? Or is this mostly a theoretical concern? Furthermore, would you attribute the significant performance gain in experiments compared to Soft TopK to the tie-breaking of the Gumbel noise? Or is it that noise+sampling
provides an effective randomization technique that enables better performance?

- Can you explain a bit the difference between soft TopK with sampling and without? Does sampling work as in the Gumbel-Sinkhorn case? I would expect
sampling to improve soft TopK as well, but it doesn't appear to be the case.

- The scope of the approach is somewhat limited as it only addresses cardinality constraints. It also seems to rely on the differentiability of the objective function.
Furthermore, conceptually this is a somewhat incremental improvement over the existing soft TopK algorithm. Therefore I am concerned that the impact of this work might be limited.

- How much does the number of samples affect performance? An ablation study on that would have been nice.


**Summary Of The Paper:**

The paper proposes an unsupervised method for cardinality-constrained combinatorial problems. By building on previous work on Soft TopK through optimal transport (OT), the authors provide an enhanced Soft TopK framework that includes Gumbel noise in the OT problem. This provides a more complete theoretical bound on the original OT problem but also strong experimental performance on two combinatorial optimization problems.

**Summary Of The Review:**

This is overall a solid paper that is theoretically motivated with strong experimental results. However, the scope of the project is somewhat limited and the proposed method is incremental in nature. Furthermore, the writing could be improved and I have some doubts/questions regarding certain experimental details. My starting grade is 6. After the authors' response, I will reconsider my evaluation.

---

> ### Author Response · Authors · 2021-11-20
> **Response to Reviewer BoAG [2/2]**
>
> **Q4: The scope of the approach is somewhat limited as it only addresses cardinality constraints. It also seems to rely on the differentiability of the objective function.  Furthermore, conceptually this is a somewhat incremental improvement over the existing soft TopK algorithm. Therefore I am concerned that the impact of this work might be limited.**
>
> - Firstly, we update the paper with Eq (1) on Page 1 to present a general form of problems that can be tackled by our proposed method:
>
>   $$ \min_{\mathbf{x}} f(\mathbf{x}) \qquad s.t. \quad ||\mathbf{x}||_0 \leq k$$
>
>   where the cardinality constraint $||\mathbf{x}||_0 \leq k$ is ubiquitous in machine learning and combinatorial optimization tasks, and our proposed method can potentially be applied to tackle a wide range of problems that involve the cardinality constraint.
>
>   Secondly, concerning the differentiability of the objective function, as far as we are able to know, for most problems it should be differentiable for $f(\mathbf{x})$ w.r.t. $\mathbf{x}$ and it may not be a major issue.
>
>   Finally, for the potential impact of this paper, we are aware that machine learning for combinatorial optimization is a trending research topic in venues like ICLR/NeurIPS/ICML, and novel machine learning tools like optimal transport (OT) and the Gumbel trick are currently under-explored in previous papers. We believe that OT and the Gumbel trick are with great potential, and we hope that this paper can inspire more researchers in developing OT and Gumbel-based combinatorial optimization learning methods in the future.
>
>
>
> **Q5: How much does the number of samples affect performance? An ablation study on that would have been nice.**
>
> - Your suggestion is appreciated and well taken and we have updated this ablation study in Table 3-5 in the supplementary material (Page 21-22). Generally, more samples (i.e. larger $\\#G$) will be more appealing, because GS-TopK will have a more accurate estimation of the objective score, and it will have a higher probability of discovering better solutions. However, $\\#G$ cannot be arbitrarily large because we are running GS-TopK in parallel on GPU, and the GPU has limited memory. Therefore, in experiments, we set a large enough $\\#G$ (e.g. $\\#G=1000$) and ensure that it can fit into the GPU memory of our workstation (2080Ti, 11G).
>
>   We post the ablation study result here for your easy reference:
>
>   |               | $\\#G=1000$ |             |             | $\\#G=800$  |                        |            |
>   | ------------- | ----------- | ----------- | ----------- | ----------- | ---------------------- | ---------- |
>   |               | $\tau$=0.01 | $\tau$=0.05 | $\tau$=0.1  | $\tau$=0.01 | $\tau$=0.05 | $\tau$=0.1 |
>   | $\sigma$=0.1  | 42513.4     | 44759.2     | **45039.5** | 42511.6     | 44754.6                | 45037.6    |
>   | $\sigma$=0.15 | 41456.5     | *44710.3*   | 44837.2     | 41421.4     | 44705.8                | 44841.5    |
>   | $\sigma$=0.2  | 41264.3     | 44638.1     | 44748.2     | 41235.9     | 44651.5                | 44748.6    |
>
>
>
> **Q6: About Minor Corrections and Typos**
>
> - Many thanks for your detailed feedback, and we have updated the paper accordingly with your generous and detailed comments:
>
>   - See Page 2 where we have updated the discussions about related works (Paulus et al., 2020; Karalias and Loukas, 2020);
>   - See Page 5 for the fixed notation $\widetilde{\mathbf{T}}^*$;
>   - See Page 5 and 7 for the fixed typos that you have mentioned.

---

> > ### Comment · Reviewer_BoAG · 2021-11-26
> > **Responding to the authors**
> >
> > The authors have provided a thoughtful response to my comments. I do not feel confident enough in the contribution of this paper to update the score to clear acceptance (8), but I do lean towards acceptance so I will maintain my original score. Using Gumbel-OT to enforce cardinality constraints is a nice combination of techniques from the literature. The authors provide a theoretical argument to back up their approach, compare with both solvers and works from the literature, and the experiments show that it works well. The authors have also put effort into fixing various typos and writing issues of the paper and have provided more experimental demonstrations in their rebuttal.
> >
> >  My concerns regarding the potential applicability/scope of the work remain. Perhaps a more diverse experimental setup apart from combinatorial optimization would have helped establish this point. I don't think this can be addressed within the context of the rebuttal though.

---

> ### Author Response · Authors · 2021-11-20
> **Response to Reviewer BoAG [1/2]**
>
> Your precious comments offer us a lot to explore, which we deeply cherish. We set out below our responses to each of the questions.
>
> **Q1: Is the theoretical result about $x_k = x_{k+1}$ only a theoretical concern, that will seldom be encountered in practice?**
>
> - We agree with the reviewer that $x_k \neq x_{k+1}$ will be more frequently encountered in practice, but the diverging gap is also critical if $x_k \neq x_{k+1}$ and $|x_k - x_{k+1}|$ is small. Theoretically, by referring to the mathematical form derived by Xie et al., 2020:
>
>   $$||\mathbf{T}^{*} - {\mathbf{T}}^{\tau *}||\_F \leq \frac{2 m\tau \log2}{|x_{k} - x_{{k+1}} |}$$
>
>   Even if $x_k \neq x_{k+1}$, given a small $|x_k - x_{k+1}|$ in the denominator, the gap will also become very large, such that SOFT-TopK (Xie et al., 2020) will fail to get a near-discrete solution which is crucial for an accurate estimation of the combinatorial objective score. In contrast, our bound has $\sigma^2$ in the denominator, thus the diverging issue will not be encountered if $\sigma$ is large enough. To better illustrate this issue, we update the toy example in Figure 1 by selecting the Top3 items from the following sequence: [1.0, 0.8, 0.601, 0.6, 0.4, 0.2], where there is a deterministic Top3 result [1.0, 0.8, 0.601]. However, as shown in the updated Figure 1, the improvement of the Gumbel trick is still significant under this scenario. (Since we cannot insert figures in openreview, please refer to Figure 1 in our updated paper.)
>
>
>
> **Q2: Would you attribute the significant performance gain in experiments compared to Soft TopK to the tie-breaking of the Gumbel noise? Or is it that noise+sampling provides an effective randomization technique that enables better performance?**
>
> - We believe the performance gain is due to the tie-breaking of the Gumbel noise, based on which our GS-TopK can offer a more accurate estimation of the combinatorial objective score. To better clarify this, some remarks are added: the self-supervised learning pipeline in our GS-TopK, SOFT-TopK (Xie et al., 2021) and EGN (Karalias & Loukas, 2020) can be viewed as optimizing the objective score of the combinatorial optimization problem, yet the discrete constraints are relaxed to enable gradient-based optimization. Since the decision variables are continuous, the estimation of the objective score may be inaccurate, due to the gap between the continuous solution and its MAP-inference discrete solution. Our theoretical analysis characterizes such a gap, and we prove that our gap improves SOFT-TopK by incorporating the Gumbel trick. To conclude, the connection between our theoretical findings and the experiment results is that our GS-TopK can estimate the objective score more accurately because our derived gap to the MAP-inference is tightened compared to SOFT-TopK. Such a connection is also validated by Figure 3 in our paper.
>
>   Besides, concerning the noise+sampling effect, we want to highlight the result of the "SOFT-TopK w/ sampling" entry. We validate by this entry that the improvement brought by GS-TopK is not simply due to the sampling effect of Gumbel noise in testing, but mainly because we can estimate the combinatorial objective score more accurately with the Gumbel noise during training.
>
>
>
> **Q3: Can you explain a bit the difference between soft TopK with sampling and without? Does sampling work as in the Gumbel-Sinkhorn case? I would expect sampling to improve soft TopK as well, but it doesn't appear to be the case.**
>
> - You get exactly the point, the sampling of "SOFT-TopK w/ sampling" works the same as in the GS-TopK, and the only difference is that "SOFT-TopK w/ sampling" performs sampling during testing, and GS-TopK performs sampling for both training and testing. As discussed above in Q2, we want to validate that the improvement brought by GS-TopK is not simply due to the sampling effect of Gumbel noise in testing, but mainly because of a more accurate estimation of the objective score during training.

---

### Author Response · Authors · 2021-11-20
**Our Response to All Reviewers [2/2]**

### We Add Ablation Study on Hyperparameters

Inspired by Reviewer BoAG, J7pw and mjiu, we conduct an ablation study about the sensitivity of hyperparameters by performing an extensive grid search near the configuration used in our max covering experiments ($\tau=0.05, \sigma=0.15, \\#G=1000$). The following ablation study results are updated in the supplementary material. We choose the k=50, m=500, n=1000 problem, and we have the following results (higher is better, we mark the configuration used in Table 1 by italic number):

**GS-TopK**

|   | $\\#G=1000$ | | | $\\#G=800$ | | |
| ----- | ------ | ---- | ---- | --- | --- | --- |
|   | $\tau$=0.01 | $\tau$=0.05 | $\tau$=0.1  | $\tau$=0.01 | $\tau$=0.05 | $\tau$=0.1  |
| $\sigma$=0.1  | 42513.4 | 44759.2  | **45039.5** | 42511.6 | 44754.6  | 45037.6 |
|  $\sigma$=0.15 | 41456.5 | *44710.3* | 44837.2 | 41421.4 | 44705.8  | 44841.5     |
|  $\sigma$=0.2 | 41264.3  | 44638.1 | 44748.2 | 41235.9 | 44651.5  | 44748.6     |

**SOFT-TopK**

| $\tau=$   | 0.001   | 0.005   | 0.01  | 0.05 | 0.1 |
| -- | --- | ---- | ---- | --- | --- |
| SOFT-TopK | 35956.6 | 42013.3 | **42520.8** | *41959.9* | 40721.17 |

**Our analysis:** under the configuration used in our paper, both SOFT-TopK and GS-TopK have relatively good results. By grid search, we also discover better hyperparameters for both GS-TopK and SOFT-TopK, but it is worth noting that SOFT-TopK is still inferior to GS-TopK with the best-discovered hyperparameters. Our grid search result shows that our GS-TopK is not very sensitive to $\sigma$ if we have $\tau=0.05$ or $0.1$, and the result of $\tau=0.01$ is inferior because the Sinkhorn algorithm may not converge. The results of $\\#G=1000$ are all better than $\\#G=800$, suggesting that a larger $\\#G$ is appealing if we have enough GPU memory. It is also discovered that SOFT-TopK seems to be able to accept a smaller value of $\tau$ compared to GS-TopK, probably because adding the Gumbel noise will increase the divergence of elements thus performs in a sense similar to decreasing $\tau$ when considering the convergence of Sinkhorn. Since we have discovered a set of better hyperparameters, we plan to validate its effectiveness on other settings and update the results in the camera-ready version accordingly.


### We Fix the Typos, Improve the Writing, and Discuss More Related Works

We truly appreciate the detailed comments from the reviewers concerning the typos, improvements of writing, and adding more related works. We have updated the paper accordingly. Please refer to our responses to each individual reviewer about the detailed revisions made as per your valuable comments.

---

### Author Response · Authors · 2021-11-20
**Our Response to All Reviewers [1/2]**

**Update after rebuttal:**

We sincerely appreciate the insightful comments and efforts from the reviewers to improve this paper, and we feel delighted that Reviewer BoAG and fJUe recognize the merits of our paper, and our rebuttal relieves the concerns from Reviewer J7pw who have raised their score. Since the discussion period is approaching its ending, we will be more than happy if Reviewer J7pw and miju could reconsider our paper if our rebuttal has addressed your concerns. Besides, we will also be truly grateful if AC could recognize our non-trivial theoretical results and strong experiment results, which are also recognized by most reviewers.

==================

We would like to express our sincere gratitude to all reviewers for their detailed feedbacks on this paper. As precisely identified by reviewers, in this paper, we develop an effective trick to incorporate the Gumbel noise with novel OT-based TopK operator (Reviewer BoAG, fJUe, mjiu), derive GS-TopK with non-trivial theoretical findings (Reviewer BoAG, fJUe, mjiu), and our performance proves to be better in the application of cardinality constrained combinatorial optimization learning (Reviewer BoAG, J7pw, fJUe, mjiu). However, there seem to be certain misunderstandings towards our paper, and here we summarize some shared issues from all the reviewers and hope our responses will address and alleviate the current concerns. Please feel free to contact us if you would like to discuss anything further and we remain available for any queries you, the program committees and public readers may have.

### We Update the Toy Example (Figure 1) as $x_k \neq x_{k+1}$

Our theoretical analysis shows that by introducing the Gumbel trick, we are capable of mitigating the diverging issue when $x_k = x_{k+1}$. Reviewer BoAG, J7pw, mjiu raise the concern about what if $x_k \neq x_{k+1}$, which will be more frequently encountered. Theoretically, by referring to the mathematical form derived by Xie et al., 2020:

$$
||\mathbf{T}^{*} - {\mathbf{T}}^{\tau *}||\_F \leq \frac{2 m\tau \log2}{|x_{k} - x_{{k+1}} |}
$$

Even if $x_k \neq x_{k+1}$, given a small $|x_k - x_{k+1}|$ in the denominator, the gap will also become very large, such that SOFT-TopK (Xie et al., 2020) will fail to get a near-discrete solution which is crucial for an accurate estimation of the combinatorial objective score. In contrast, our bound has $\sigma^2$ in the denominator, thus the diverging issue will not be encountered if $\sigma$ is large enough. To better illustrate this issue, we update the toy example in Figure 1 by selecting the Top3 items from the following sequence: [1.0, 0.8, 0.601, 0.6, 0.4, 0.2], where there is a deterministic Top3 result [1.0, 0.8, 0.601]. However, as shown in the updated Figure 1, the improvement of the Gumbel trick is still significant under this scenario. (Since we cannot insert figures in openreview, please refer to Figure 1 in our updated paper.)



### About the Connection between Our Theoretical Findings and the Experiment Results

A thought-provoking question is raised by Reviewer BoAG, fJUe that why our theoretical result will affect the performance of combinatorial optimization learning. Our answer is: with the tie-breaking of Gumbel noise in our theoretical findings, our GS-TopK can offer a more accurate estimation of the combinatorial objective score. The self-supervised learning pipeline in our GS-TopK, SOFT-TopK (Xie et al., 2021), and EGN (Karalias & Loukas, 2020) can be viewed as optimizing the objective score of the combinatorial optimization problem, yet the discrete constraints are relaxed to enable gradient-based optimization. Since the decision variables are continuous, the estimation of the objective score may be inaccurate, due to the gap between the continuous solution and its MAP-inference discrete solution. Our theoretical analysis characterizes such a gap, and we prove that our gap improves SOFT-TopK by incorporating the Gumbel trick. To conclude, the connection between our theoretical findings and the experiment results is that our GS-TopK can estimate the objective score more accurately because our derived gap to the MAP-inference is tightened compared to SOFT-TopK. Such a connection is also validated by Figure 3 in our paper.

### We Compute Optimal Values by Gurobi

Based on the precious suggestions by Reviewer fJUe, we update Table 2 with the optimal objective scores and the gap to optimal objectives for the discrete clustering problem. Our method takes less than 1/3 time cost compared to the optimal Gurobi solvers, and the approximation ratios are around 2%. The speedup is more significant for the larger-sized problems. We also try to acquire the optimal result by Gurobi for the max covering problem, however, the solver does not reach the optimal solution within 24 hours for a single problem instance thus we report an out-of-time (OOT) in Table 1.

---

### Decision · Program_Chairs · 2022-01-20

**Decision:**

Reject

**Comment:**

The authors propose a random perturbation on top of a soft top-k operator that builds upon entropic regularized optimal transport (when applied to a 1D problem). The motivation of the paper is built around an approximation bound (proposed in the Xie et al '20 paper) that compares the true OT matrix from the regularized OT matrix in the case where some of the 1D entries from which one wishes to extract top-k values are very close (eg. x_{t} ~ x_{t+1}). The authors argue that this bound, with inverse dependencies in the closest element in the list, diverges.

The authors state that this possible divergence is an issue, because values to be sorted/top-ked can be very close in practice. To solve this issue, the authors introduce instead a Gumbel noise mechanism that no longer makes the bound diverge, through a fairly long theoretical analysis. The approach now requires the recomputation for several noisy inputs of the same regularized OT estimator. The authors propose then to use these soft-top-k approaches to solve a combinatorial problem using gradient descent, namely a capacity constrained problem and clustering, including some tricks on controlling both entropy regularization and Gumbel noise magnitude.

The paper has generated a long discussion among the AC and reviewers. While the paper has a few strong points that were appreciated (interest of empirical validation which seems to suggest some improvements over commercial solvers on considered setups), there remain a few issues.

The theoretical side of the paper is bit blurry. The idea of introducing Gumbel noise on top of an already soft operator is not completely clear, since these perturbations are there to add differentiability to something (reg-OT) that was introduced itself to be differentiable. The theoretical motivation is unclear: the noise is introduced because the _upper bound_ diverges (and not the gap between the "true" OT and entropic OT, since it is always bounded). The perturbation mechanism is only motivated to improve the limitations of an upper bound, not of the original algorithm itself. What's more, it's not entirely clear why that gap should be decreased (between true and regularized OT) since it has to exist to obtain some differentiability. While the study of the gap itself was added during the discussion phase in Fig. 1"A toy example to explain Lemma 2", one would expect better foundations for this idea.

With a somewhat unclear theoretical motivation, the experiments should be very convincing. Reviewers have noted some issues related to comparing CPU/GPU times. While I am sympathetic to the problems encountered by the authors when running such comparisons, these issues should be properly reflected in their initial claims, and not appear in the rebuttal only. I also think experiments are still lacking in diversity. For instance, the k-means problem is studied in 2D (begging the natural question of whether such an improvement would remain in higher dimensions). I could not find a clear statement on the number of repeats carried out to obtain error bars. Since I don't envision either of the max-covering problem nor k-means to become the "killer app" of this paper, I would encourage the authors to consider problems that are less synthetic.